# Is deoxygenation detectable before warming in the thermocline?

Angélique Hameau[1, 2], Thomas L. Frölicher[1, 2], Juliette Mignot[3], and Fortunat Joos[1, 2]

[1]Climate and Environmental Physics, Physics Institute, University of Bern, Switzerland
[2]Oeschger Centre for Climate Change Research, Bern, Switzerland
[3]LOCEAN/IPSL, Sorbonne Université (SU)-CNRS-IRD-MNHN, Paris, France

**Correspondence:** Angélique Hameau (hameau@climate.unibe.ch)

**Abstract.** Anthropogenic greenhouse gas emissions cause ocean warming and oxygen depletion, with adverse impacts on marine organisms and ecosystems. Warming is one of the main indicators of anthropogenic climate change, but, in the thermocline, changes in oxygen and other biogeochemical tracers may emerge from the bounds of natural variability prior to warming. Here, we assess the Time of Emergence (ToE) of anthropogenic change in thermocline temperature and thermocline oxygen within an ensemble of Earth system model simulations from the fifth phase of the Coupled Model Intercomparison Project. Changes in temperature typically emerge from internal variability prior to changes in oxygen. However, in about a third (35 $\pm11$ %) of the global thermocline deoxygenation emerges prior to warming. In these regions, both reduced ventilation and reduced solubility add to the oxygen decline. In addition, reduced ventilation slows the propagation of anthropogenic warming from the surface into the ocean interior, further contributing to the delayed emergence of warming compared to deoxygenation. Magnitudes of internal variability and of anthropogenic change, which determine ToE, vary considerably among models leading to model-model differences in ToE. We introduce a new metric, relative ToE, to facilitate the multi-model assessment of ToE. This reduces the inter-model spread compared to the traditionally evaluated absolute ToE. Our results underline the importance of an ocean biogeochemical observing system and that the detection of anthropogenic impacts becomes more likely when using multi-tracer observations.

# 1 Introduction

Carbon emissions from human activities are causing ocean warming (Rhein et al., 2013) and ocean deoxygenation, i.e. a decrease in the oceanic oxygen ($O_2$) concentration (Sarmiento et al., 1998; Bopp et al., 2002; Matear and Hirst, 2003; Battaglia and Joos, 2018). Both warming and deoxygenation adversely affect marine organisms, ecosystems and the services they provide (e.g. Pörtner et al., 2014; Deutsch et al., 2015; Gattuso et al., 2015; Magnan et al., 2016).

All major ocean basins have experienced significant warming over the last few decades. Warming is generally strongest at the surface and weaker at deeper layers, indicative of heat penetrating from the surface towards the deep ocean as expected from atmospheric greenhouse gas forcing. The strongest warming in the top 2000 m has been observed in the Southern Ocean (Roemmich et al., 2015) and the tropical/subtropical Pacific and Atlantic Ocean (Cheng et al., 2017). On regional to local scales, the anthropogenic warming signal may be masked by natural interannual to multi-decadal variability. For example, decadal-scale cooling trends in the tropical Pacific and Indian oceans may arise from natural El Niño-Southern Oscillation and/or Indian Ocean Dipole variability (Han et al., 2014). Similarly, decadal variability in the Atlantic Meridional Overturning is observed to modulate temperature and heat content change in the North Atlantic (Chen and Tung, 2018).

Observation-based studies indicate that the global ocean oxygen content has decreased since 1960 (e.g. Schmidtko et al., 2017). Increased ocean surface temperature reduces oxygen solubility, limiting atmospheric oxygen dissolution into the upper ocean. Increased surface temperature reduces oxygen solubility, limiting atmospheric oxygen dissolution into the upper ocean. In subsurface waters, oxygen concentration is also affected by changes in ventilation and the remineralisation of organic matter. In the contemporary ocean, oxygen decreases in the interior are mostly dominated by a reduction in ventilation with a smaller role for changes related to the production of organic matter, $O_2$ solubility, and air-sea equilibration of $O_2$ in surface waters (Bopp et al., 2002; Plattner et al., 2002; Bopp et al., 2017; Tjiputra et al., 2018; Hameau et al., 2019). The largest oxygen declines are located in the Pacific Ocean (equator and northern hemisphere) and the Southern Ocean. However, observations are relatively sparse and only start in the second half of the 20th century. Therefore, it is challenging to distinguish human-caused trends from natural variations in the observational record of ocean $O_2$.

Global climate models, such as the Earth system models that participated in phase 5 of the Coupled Model Intercomparison Project (CMIP5) reproduce the long-term trend in global ocean heat content over the last 50 years when uncertainties of observation-based estimate and internally generated natural variability are taken into account (Frölicher and Paynter, 2015; Cheng et al., 2019). Modelling studies agree on the sign of oceanic $O_2$ changes, but likely underestimate the magnitude of loss (Bopp et al., 2013; Cocco et al., 2013; Oschlies et al., 2017). In particular in the tropical regions, models are not able to reproduce observed $O_2$ decrease in equatorial low-oxygen zones (Stramma et al., 2008; Cocco et al., 2013; Cabré et al., 2015).

It is expected that ocean warming and deoxygenation, and the combination thereof, increases the risk of adverse impacts on marine organisms and ecosystem services (Pörtner et al., 2014). Warming of the ocean influences the physiology and ecology of almost all marine organisms. Reduced oceanic $O_2$ concentrations can disrupt marine ecosystems by pushing organisms to their species-specific limits of hypoxic tolerance, below which the species are no longer able to meet their metabolic $O_2$ demand. The species-specific metabolic demand of $O_2$ is also a function of temperature, as warmer temperatures increase

metabolic rates and oxygen requirements (Deutsch et al., 2015). At the same time, higher ocean temperatures also decrease oxygen supply through reduced ventilation, enlarging the regions with limited $O_2$ concentrations and thus shifting ecosystem distribution (Cheung et al., 2011).

Beyond the combined impact of physical and biogeochemical changes, an interesting question is whether anthropogenic changes in the ocean interior are first detectable in variables that are routinely and frequently measured such as temperature (T) or in variables with a relatively low observational coverage but potentially high impact for ecosystems such as $O_2$ (Joos et al., 2003). The answer may have implications for measurement strategies to detect anthropogenic changes in subsurface waters as well as for the impacts of physical and biogeochemical change on marine life. For the surface ocean, earlier studies (Keller et al., 2014; Rodgers et al., 2015; Frölicher et al., 2016; Schlunegger et al., 2019) showed that the anthropogenic signals of pH and pCO2 emerge earlier than sea surface temperature and $O_2$ change and earlier than productivity changes. Changes in surface $O_2$ are tightly coupled to temperature-driven solubility changes and $O_2$ varies hand in hand with sea surface temperature and the two signals emerge typically concomitantly. Regarding the ocean interior, the sequence of emergence for $O_2$ and T is less clear. Global warming increases surface ocean temperature, which tends to reduces $O_2$. On the other hand, $O_2$ is also influenced by non-thermal processes, such as respiration and the redistribution by ocean circulation and mixing. Respiration of organic matter in the ocean interior may have a larger influence on $O_2$ change than temperature-driven solubility change in a more stratified and less ventilated ocean. One could therefore expect that, under global warming, the combined effect of increased $O_2$ consumption and decreased $O_2$ solubility will accelerate the $O_2$ depletion in subsurface waters and that $O_2$ may be detectable before the warming reaches that layer.

The concept of Time of Emergence (ToE; Christensen et al., 2007; Hawkins and Sutton, 2012) is often used to determine the point in time when the anthropogenic signal becomes larger than the range of natural variability. ToE has been broadly used in climate change detection for physical climate variables (e.g. surface temperature: Hawkins and Sutton, 2012; Frame et al., 2017), land carbon fluxes (Lombardozzi et al., 2014) or marine biogeochemical variables (e.g. pH, alkalinity, DIC, pCO$_2$: Hauri et al., 2013; Keller et al., 2014; marine biological productivity: Henson et al., 2016). A limited number of studies addressed anthropogenic deoxygenation detection in the subsurface layers (Rodgers et al., 2015; Frölicher et al., 2016; Henson et al., 2016; Long et al., 2016; Henson et al., 2017; Hameau et al., 2019). One study, Hameau et al. (2019), uses a single model (CESM), to investigate ToE of temperature and oxygen in the thermocline, finding that anthropogenic ocean warming emerges much earlier than the $O_2$ signal in low and midlatitude regions. Delayed emergence of changes in $O_2$ is due to the opposing effects of $O_2$ solubility and $O_2$ consumption. In the high latitudes and the Pacific subtropical gyres, deoxygenation emerges before ocean warming in CESM. This occurs because decrease in oxygen solubility are reinforced by increased $O_2$ consumption, leading to strong $O_2$ depletion. However, it is unknown if this single-model result is robust across a suite of different Earth system model simulations. Here, we conduct a multi-model study to more broadly test the hypothesis that anthropogenic deoxygenation in the thermocline emerges prior to anthropogenic warming. Since the primary objective is to test the consistency across models of the order of emergence (deoxygenation prior to warming) within a single model. We introduce and use a relative ToE to conduct the intercomparison, rather than the absolute year of ToE. We define relative ToE as a deviation relative to the model mean ToE for improved model intercomparison.

In this study, we analyse and compare the relative ToE(T) and ToE(O$_2$) in the thermocline (200 – 600 m) using nine different CMIP5 Earth system models. We also assess the impact of using the relative ToE in comparison to the classical approach using absolute ToE. In addition, we discuss the magnitude of background internal variability and anthropogenic signal, and their translation into ToE. Finally, we analyse the role of solubility, ventilation and respiration for the emergence of anthropogenic changes in oxygen and temperature.

## 2  Method

### 2.1  Earth system models

We use output from eight different configurations of four Earth system models (ESMs) that participated in the Coupled Model Intercomparison Project 5 (CMIP5; Taylor et al., 2012): GFDL-ESM2M, GFDL-ESM2G, HadGEM2-CC, IPSL-CM5A-LR,

IPSL-CM5A-MR, IPSL-CM5B-LR, MPI-ESM-LR and MPI-ESM-MR (Table 1). In order to extent the multi-model ensemble from four to five family-models, we also included the output from simulations performed with the Community Earth System Model (CESM1.0) conducted at the Swiss Supercomputing Centre. The horizontal ocean model resolution is about 1° in both the GFDL models and CESM1.0. HadGEM2-CC and IPSL models have a horizontal resolution of about 2° and the MPI models have a horizontal resolution of about 0.4° (MR) and 1.5° (LR). Of the nine models, all but one (GFDL-ESM2G, isopycnal

vertical coordinate) use a pressure-based vertical coordinate. For additional information on the individual model setups, the reader is referred to the references listed in Table 1.

Both the CMIP5 ESMs and the CESM1.0 were run under prescribed anthropogenic and natural greenhouse gas and aerosol forcing. All simulations span the historical 1861-2005 period and the 2006-2100 period following the Representative Con-

centration Pathway 8.5 (RCP8.5) scenario. The RCP8.5 represents a high emission scenario with a radiative forcing of 8.5 W m$^{-2}$ in year 2100 (Riahi et al., 2011). These simulations are complemented with output from corresponding control runs with constant preindustrial forcing. The CESM1.0 simulations differ from the CMIP5 simulations only with regard to the spin-up procedure: The CMIP5 model simulations are branched off from preindustrial control simulations, whereas the CESM1.0 simulation is an extension of a last millennium simulation run under 850 CE conditions (Lehner et al., 2015). For this study, all

CMIP5 models are used for which the 3 dimensional output of oxygen, temperature and salinity for all simulations were available on the Earth System Grid. We regridded all model output onto a regular 1° x 1° grid. Even though the model drift in the control simulations is relatively small in the thermocline (3.6±2.4 x 10$^{-3}$ mmol m$^{-3}$ year$^{-1}$ for trend in global mean oxygen concentration and 7.2±6.6 x 10$^{-5}$ °C year$^{-1}$ for trend in global mean temperature averaged over 200 – 600 meters), we detrended all model output with a linear trend obtained from the preindustrial control simulation in each grid cell. The CESM1.0

simulation also shows some model drift.Therefore, an exponential curve was fitted to the annual output of its associated control simulation at each grid cell. The detrending procedure is described in detail in Hameau et al. (2019).

### 2.2  Multi-model analysis methods

We use the concept of Time of Emergence (ToE; e.g. Hawkins and Sutton 2012) to compare anthropogenic changes in $O_2$ and temperature (signal; $S$) with internal variations (background noise; $N$). Here, ToE represents the moment in time at which the

ocean state becomes distinct from the preindustrial state. Appendix Figure A1 provides a graphical illustration of the method used to compute ToE.

We define the absolute ToE as the first year when the anthropogenic signal $S$ becomes equal or larger than twice the noise of internal variability $N$ (Eq. 1; following Hameau et al., 2019; Fig. A1). The threshold is set to two in order to distinguish the signal from the noise at 95 % confidence level. Annual $O_2$ and T data are first averaged over the thermocline (200 – 600 m) at each grid point of the horizontal grid and local $S$ and $N$ are computed from these depth-averaged values for each model, variable and (horizontal) grid-point. Annual anomalies are calculated relative to the preindustrial period (1860 – 1959).

$$\text{ToE} : \frac{S}{N} \geqslant 2 \tag{1}$$

The background noise, $N$, is computed as one standard deviation (SD) of $O_2$ and of T from the annual preindustrial control output. The entire duration of the control simulation is considered for each model to estimate the background noise. $N$ represents the noise due to the internal chaotic variability of the climate system. Note that this definition of the noise differs from Hameau et al. (2019), who used internal plus externally-forced natural variability from a last millennium simulation to assess the standard background noise.

The annual output of the forced, transient simulation (1860 – 2099) is smoothed by a low pass spline filter (Enting, 1987) to estimate $S$ for each (horizontal) grid point in the thermocline. The cut-off period of the spline is set to 80 years to remove decadal to multi-decadal variations (e.g. associated with internal variability). The signal $S$ is then defined as the value of the spline at each point in time. To ensure that $S$ indeed detects anthropogenic trend, we also apply a criterion for the sign of $S$ to define ToE: $S$ needs to have the same sign as the difference between the last 30 years of the future simulation and the preindustrial average for the corresponding variable and grid point.

In order to minimise inter-model differences and to highlight the common spatial patterns of ToE, we introduce a new metric, the relative ToE (ToE$_{rel}$). It is defined as the absolute ToE (ToE$_{abs}$) minus the global area-averaged ToE (ToE$_{glob}$; Eq. 2).

$$\text{ToE}_{rel} = \text{ToE}_{abs} - \text{ToE}_{glob} \tag{2}$$

$S$, $N$, ToE and ToE$_{rel}$ are first computed from the annual output for each model and at each (horizontal) grid cell. Then, multi-model median and spread (interquartile range) of the multi-model estimations are computed from the model ensemble. The median represents a "best" estimate and the interquartile range a measure of model uncertainty. Uniform weights are applied to each model configuration to compute those statistics. Tests have been performed using a weighted median as several simulations stem from the same model family (CESM x 1; GFDL x 0.5; HadGEM2 x 1; IPSL x 0.3; MPI x 0.5). However, median and interquartile range of the multi-model ensemble are not sensitive to the weighting scheme applied (not shown). Because an anthropogenic signal may not emerge before the end of the simulation in year 2100, ToE can be undefined. We therefore require that ToE values are defined for at least seven out of nine models to compute the multi-model statistics (median and spread). If more than two models have an undefined ToE, we mask the grid points in maps of the multi-model median and of the multi-model the spread.

## 2.3 Separating mechanisms of oxygen change

To diagnose processes driving the simulated changes in ocean $O_2$, the direct thermal/solubility component of change ($O_{2,sol}$) can be isolated from the total $O_2$ change. The residual, Apparent Oxygen Utilisation (AOU), represents the summation of all non-thermal changes, including those resulting from changes in ventilation and remineralisation.

$$[O_2] = [O_{2,\,sol}] + [\text{-AOU}] \tag{3}$$

The solubility component for each model is computed following Garcia and Gordon (1992), which requires local salinity and temperature output. The solubility depends mostly on temperature with a small contribution of salinity. The non-thermal component ([-AOU]) is deduced from the difference between $O_{2,sol}$ and $O_2$ following Eq. 3. In Sect. 3.4, we will use changes in [-AOU] as a proxy for changes in water mass age and ventilation. Output of an ideal age tracer is not available for most models. A decrease in water exchange between the surface ocean and the thermocline typically leads to an increase in water mass age in the thermocline. Therefore, changes in ventilation affect the balance between the rate of supply of $O_2$-rich waters from the surface and the rate of $O_2$ consumption by remineralisation of organic matter. It has been demonstrated in earlier studies (e.g. Gnanadesikan et al., 2012; Bopp et al., 2017; Hameau et al., 2019) that a decrease in [-AOU] typically corresponds to a decrease in ventilation and an increase in water mass age, as simulated changes in the remineralisation rates of organic material and in associated $O_2$ consumption are relatively small over the 21st century.

## 3 Results

### 3.1 Relative Time of Emergence

We start by discussing the multi-model median and spread of relative ToE estimates for potential temperature (Fig. 1a, b) and dissolved oxygen (Fig. 1d, e) changes in the thermocline (200 – 600m). An analysis of the roles of internal variability and anthropogenic change ToE and why anthropogenic change is detectable early or late is presented in Sect. 3.3.

### 3.1.1 Anthropogenic warming

$ToE_{rel}(T)$ shows early emergence in low latitudes and between 30° S and 60° S, and late emergence in the western tropical Pacific, in the Atlantic subpolar gyre and the subtropical gyres of the Indian and Pacific Ocean (Fig. 1a). The northern Indian Ocean and the eastern equatorial Atlantic stand out as the regions with earliest emergence in anthropogenic warming, i.e. 70 years (median of nine $ToE_{rel}(T)$) before the global average ToE. No emergence of warming by the end of the 21st century (for at least 3 models; cf. Sect. 2.2) is simulated in the subtropical gyres of the Indian and the Pacific oceans, south of Greenland and locally south of 60° S.

The multi-model spread in $ToE_{rel}(T)$ is generally small in regions with early emergence (Fig. 1b). This is the case in many regions of the Pacific and the Southern Ocean ($\pm15$ years). However, in the Atlantic subtropical gyres and in the Arabian Sea, the early $ToE_{rel}(T)$ estimates are associated with a wider spread across models ($\pm25$ to $\pm45$ years). Large inter-model spread is also found in the Kuroshio extension and in the Indian and Atlantic region of the Southern Ocean ($\pm50$ years). On average, the multi-model spread for $ToE_{rel}(T)$ is about 25 years.

ToE values for individual horizontal grid cells are globally averaged to obtain an area-weighted global mean ToE for the thermocline and each model. These global mean values range between year 1963 and year 2033 for the nine models (see subtitles in Fig. 2). The patterns of $ToE_{rel}(T)$ for each individual model are shown in Fig. 2. As described previously, low latitude regions and parts of the Southern Ocean show earlier emergence compared to mid- and other high-latitude regions. The HadGEM2-CC model (Fig. 2c) is an exception in that respect as temperature emerges later (+30 to +50 years) than the global average in the tropical Atlantic and Pacific. In the Pacific and Indian subtropical gyre regions, the models show late (IPSL family) or no emergence. And finally, CESM and the IPSL family models are the only models that show emergence before the end of the 21st century in the subtropical gyres of the Pacific.

### 3.1.2 Anthropogenic deoxygenation

In contrast to $ToE_{rel}(T)$, most of the thermocline shows no emergence of the anthropogenic $O_2$ change by the end of the 21st century (Fig. 1d). In the remaining regions $ToE_{rel}(O_2)$ varies by about $\pm40$ years. Early emergence is found in the subtropical

gyre of the North Pacific, the northern North Atlantic, the Atlantic sector of the Southern Ocean, and generally south of 60° S. No emergence is simulated in 47 % of the ocean area by the end of the 21st century including large parts of the tropics and the subtropical gyres of the Atlantic Ocean and the Indian Ocean.

5  The multi-model spread for $ToE_{rel}(O_2)$ is 20 years in the global average and thus somewhat smaller than for $ToE_{rel}(T)$. The models show a high spread for $ToE_{rel}(O_2)$ ($\pm 50$ years) at low latitudes, such as in the southern Arabian Sea or in the equatorial Atlantic, whereas high model agreement is found in parts of the central North Pacific and the northern Indian Ocean (spread of $\pm 15$ years) (Fig. 1e). In the eastern tropical Atlantic, the spread for $ToE_{rel}(O_2)$ is, despite a smaller global mean spread, larger than for $ToE_{rel}(T)$. In summary, even though the median pattern of $ToE_{rel}(O_2)$ is relatively uniform in comparison to

10 $ToE_{rel}(T)$, the spread for $ToE_{rel}(O_2)$ varies between regions as for $ToE_{rel}(T)$.

  The multi-model median $O_2$ signal does not emerge in 47 % of the global thermocline as noted above. Mid and low latitudes show no emergence by the end of the 21st century in most of the models (Fig. 3). However, the exact regions of no emergence differ between models. This regional mismatch, in combination with the requirement that at least seven out of nine models

15 need to show an emerging signal (Sect. 2.2), explains why in the multi-model analysis many grid cells are masked, indicating no emergence in the median (Fig. 1d-f). The area fraction with no emerging $O_2$ signal is smaller in individual models than in the multi-model median and ranges between 10 and 30 %.

  As for temperature, a large range in absolute ToE is found with globally-averaged $ToE(O_2)$ ranging between the year 1991

20 and 2046 for the nine models (see subtitles in Fig. 3). The analysis of $ToE_{rel}(O_2)$ for individual models reveals some additional notable differences (Fig. 3). GFDL-ESM2M, GFDL-ESM2G, HadGEM2-CC and CESM1.0 simulate early emergence in the Southern Ocean, but the IPSL models project no emergence of deoxygenation in this region by the end of the 21st century. In addition, the IPSL models and the CESM1.0 model show relatively early emergence in many grid cells of the western tropical Pacific, a region with no emergence in other models. $ToE_{rel}(O_2)$ also diverges across the models in the Atlantic subtropical

25 gyres: in the HadGEM2 and IPSL simulations, oxygen changes are simulated to emerge relatively early ($ToE_{rel}(O_2) \sim 40$ to 60 years), whereas in the GFDL, MPI and CESM simulations, the changes are not yet detectable by the end of the 21st century.

### 3.2 Relative versus absolute ToE

Mapping $ToE_{rel}$ for different models is intended to emphasise common patterns across models by removing the global mean bias between models, while model-model differences in $ToE_{abs}$ are indicative of an overall model uncertainty.

30 The multi-model spread for $ToE_{abs}$ is in average larger than the multi-model spread for $ToE_{rel}$ for temperature (Fig. 1c) and oxygen (Fig. 1f), while spatial patterns are similar for $ToE_{rel}$ and $ToE_{abs}$. On global average, the spread is reduced from $\pm 30$ years for $ToE_{abs}(T)$ to $\pm 23$ years for $ToE_{rel}(T)$ and from $\pm 20$ years for $ToE_{abs}(O_2)$ to $\pm 17$ years for $ToE_{rel}(O_2)$. Regionally, the reduction can be larger. For example, in the equatorial regions, the Atlantic and the Southern Ocean, the spread is reduced by 20 to 50 years when computed for $ToE_{rel}(T)$ instead for $ToE_{abs}(T)$. Similarly, the spread in $ToE(O_2)$ is reduced from $\pm 35$ to

±5 years in parts the North Pacific.

## 3.3 Internal variability and anthropogenic signals

The ToE allows for a comparison across climate models, by combining the amplitude of the climate response to anthropogenic forcing and the amplitude of natural variability in one metric. The magnitude and the spatial patterns of the internal variability and of the anthropogenic signal for both thermocline temperature and oxygen are discussed next.

The multi-model median of internal variability for thermocline temperature fluctuates with an amplitude typically ranging between ±0.1 °C in the tropics and the Arctic Ocean, and ±0.5 °C in mid-to-high latitudes (Fig. 4a). SD(T) is the largest (up to ±0.9 °C) in the Western Boundary Currents such as the Kuroshio Current and the Gulf Stream. The internally generated variability is also relatively large along the equatorward flanks of the subtropical gyres. It is also in these regions where SD(T) differs most among models (up to ±0.5 °C along the North Atlantic Current; Fig. 4c).

In the multi-model median, temperature in the thermocline is projected to increase on global average by 1.2±0.7 °C (Fig. 4b) by the end of the 21st century under the RCP8.5 scenario relative to the period 1861-1959, in accordance with (Levitus et al., 2009, 2012; Bilbao et al., 2019). Large warming of more than 4.0±0.7 °C is projected in the northern North Atlantic and around the subantarctic water in the Indian and Atlantic Oceans (Fig. 4b and Fig. S4). We note that these projections are also characterised with the largest inter-model spread (±1.5 °C; Fig. 4d and Fig. S4) and uncertainties in these regional warming projections are large . Finally, disagreement among models in simulating changes in thermocline temperature is also large in the Arctic Ocean, possibly related to different simulated changes in sea ice cover (Stroeve et al., 2012; Wang and Overland, 2012).

The combination of a strong signal and small variability results in early detection of the changes. This is the case in the Southern Ocean at 45° S (in the Atlantic and Indian regions; Fig. 1a), where the anthropogenic warming is strong (up to 4 °C; Fig. 4b) but the internal variability is relatively small (0.1 °C to 0.3 °C; Fig. 4a). However, early emergence of anthropogenic changes can also occur when the signal is relatively small, if the variability is even smaller. This is the case in the tropical oceans such as in the Arabian Sea, the equatorial Atlantic and the western equatorial Pacific, where water masses warm modestly (up to 1.5 °C), but vary naturally between 0.1 °C and 0.2 °C only. It is also the case in the eastern equatorial Pacific, where the early emergence arise from the very weak internal variability in the thermocline, although, the temperature increase (∼0.80 °C) is also relatively weak. In this region, the substantial variability in $O_2$ and T is largely confined to the top 200 m. No emergence by the end of the 21st century, such as simulated in the subtropical gyres of the Indian and Pacific oceans, results from a relatively weak signal combined with a relatively strong variability in these regions.

Internal variability of dissolved oxygen concentrations is particularly large in the northern North Pacific and North Atlantic, the Southern Ocean and along the equatorward boundaries of the subtropical gyres with $SD(O_2)$ of up to 10 mmol m$^{-3}$ (Fig. 5a). The multi-model spread of $SD(O_2)$ (Fig. 5c) is about equally large as the median of $SD(O_2)$ (Fig. 5a) along the equatorward boundaries of the subtropical gyres. Looking at the individual model responses, the $O_2$ internal variability shows a wide range of different patterns (Fig. S5). The GFDL and MPI models simulate high internal variability of oxygen in the entire thermocline, whereas CESM, HadGEM2 and IPSL models show high variability regionally.

The $O_2$ concentration in the thermocline (Fig. 5b) is projected to decrease under global warming, in accordance with previous model studies (e.g. Sarmiento et al., 1998; Cocco et al., 2013; Bopp et al., 2017). The anthropogenic decrease in $O_2$ is large in the Southern Ocean, in the North Pacific subtropical gyre and in the North Atlantic subpolar gyre. In tropical regions, the changes are projected to be small, except for the western Indian ocean, where more than 70 % of the models project an increase of $O_2$ concentration. The simulated $O_2$ changes differ most across models in high latitudes and in the subopolar gyres, as well as in the equatorial Indian ocean (Fig. 5d).

Despite differences in the simulated magnitude of $O_2$ changes and internal variability patterns of $O_2$ between the different models, the resulting ToE$_{rel}$($O_2$) are robust across models. For example, the decrease in $O_2$ spans from -12 to -40 mmol m$^{-3}$ (Fig. S6) and $SD(O_2)$ spans from $\pm5$ to $\pm15$ mmol m$^{-3}$ (Fig. S5) in the central North Pacific. Moreover, the spatial locations of the maximum $O_2$ depletion differ across the models. However, ToE$_{rel}$($O_2$) in this region is within $\sim$10 years (Fig. 1d), with a relatively low spread ($\pm10$ years) compared to ToE$_{abs}$($O_2$) ($\pm30$ years). Another example is the CESM model. The very early detection of anthropogenic changes (for temperature and oxygen) in the CESM model described in Sect. 3.1, results from a particularly weak internal variability (Figs. S3i and S5i; see also Hameau et al., 2019) combined with a high climate sensitivity of the model (Figs. S4i and S6i). The ToE$_{rel}$ allows the comparison of ToE resulting from CESM output with the results from the 8 models in spite of these model-model differences (Figs. 2 and 3).

## 3.4 Comparison of ToE($O_2$) with ToE(T)

In general, temperature changes are detectable before $O_2$ changes in around 64$\pm$11 % of the thermocline (yellow to brown colours in Fig. 7). As discussed in section 3.1, the anthropogenic $O_2$ signal emerges late or not at all in many low latitude regions, while the anthropogenic warming signal is emerging in most regions and typically early around the equator. However, there are also areas where anthropogenic deoxygenation is detectable earlier than anthropogenic warming in all models (green to blue colours in Fig. 7). These cover 35$\pm$11 % of the global thermocline in the nine models. They are mainly located in the mid latitudes, especially between $\sim$15° N and 30° N in the North Pacific, around Antarctica (including the Ross and Weddell Sea), along the Western Australian Current and the Pacific southern subtropical gyre region. Model results for the Atlantic subtropical gyres are mixed. Some models suggest $O_2$ changes to be detectable earlier than T changes (HadGEM2 and the

IPSL family), whereas in other models the $O_2$ signal does not even emerge.

The exact locations of relatively early emergence of $O_2$ differ across models. Hence, the regions where at least seven out of the nine models show consistently an earlier emergence of $O_2$ than T is smaller and amounts to 17 % of the global thermocline area. As shown in Fig. 6 (blue areas) the $O_2$ signal emerges consistently in at least seven models before the T signal in parts of the Pacific subtropical gyres, the Southern Ocean and the southeast Indian Ocean.

A mechanistic explanation of early or late emergence of the $O_2$ signal relative to the temperature signal is not straightforward as two ratios ($S/N$) are involved. Nevertheless, changes in apparent oxygen utilisation ($\Delta$[-AOU]; Fig. 8) provide some insight into underlying mechanisms. We use $\Delta$[-AOU] as a proxy for changes in water mass age and ventilation as noted in Sect 2.2.

Regions with early emergence of anthropogenic $O_2$ compared to T show typically a decrease in [-AOU] (Fig. 7 versus Fig. 8), whereas regions with early emergence of T compared to $O_2$ show typically an increase in [-AOU]. For example, [-AOU] is decreasing in 77$\pm$8 % of the areas with early emergence of $O_2$, while only 22$\pm$8 % of these regions show an increase in [-AOU] (Fig. 9; blue). In most regions where T is emerging before $O_2$ (Fig. 9; brown), [-AOU] is increasing (62$\pm$12 %). A decreasing trend in [-AOU] is indicative of a reduced ventilation induced by upper ocean warming and increased stratification (e.g. Capotondi et al., 2012). A more sluggish ventilation slows the supply of $O_2$ from the surface to the ocean interior. Consequently, thermocline [$O_2$] and [-AOU] are both decreasing. This leads to a strong and thus early detectable anthropogenic deoxygenation. In addition, a more sluggish ventilation slows the penetration of the anthropogenic warming signal from the surface to the interior, and similarly the penetration of the thermally driven $O_2$ signal ([$O_{2,sol}$]). The detection of the temperature changes is thus delayed compared to AOU and $O_2$. There are some exceptions to this relationship between [-AOU] and the earlier emergence of $O_2$ than T. For example, $O_2$ change emerges before warming in the GFDL model around 30° S and 120° W, although [-AOU] is increasing in this region. However, warming is emerging very late as the GFDL models simulate weak warming and even some cooling (Fig. S4) in this part of the thermocline. Thus, in this special case, the early emergence of $O_2$ relative to T is due to the absence of large warming in a region with notable temperature internal variability.

Regions where the warming signal is detectable before the deoxygenation are typically associated with an increase in [-AOU]. Such increase counteracts the decrease in [$O_{2,sol}$], leading to relatively smaller changes in [$O_2$], which are thus often not detectable. There are again a few exceptions. For example, the IPSL models simulate a decrease in [-AOU] in the northern North Pacific, but an earlier ToE for T than for $O_2$ in this region.

In summary, anthropogenic change in temperature is detectable earlier than anthropogenic change in $O_2$ in most of the global ocean. However, there are large ocean regions where anthropogenic $O_2$ changes are detectable earlier in the thermocline in all models. Early emergence of deoxygenation relative to warming is typically detected in regions where thermocline ventilation and [-AOU] are decreasing over the simulation and late emergence of $O_2$ changes where ventilation and [-AOU] are increasing.

## 4   Discussion and conclusions

We analysed the time of emergence (ToE) of human-induced changes in oxygen ($O_2$) concentrations and temperature (T) in the thermocline (200 – 600 m) using nine Earth system model simulations of the climate over the historical and the future period. Using ToE as a metric allows for the assessment of anthropogenic changes by comparing the magnitude of the human-induced changes with the magnitude of internal variability. Both the magnitude of anthropogenic change and internal variability are model dependent, rendering the absolute year of ToE strongly model-dependent. Evaluating differences in absolute year of ToE, however, can obscure important model agreement upon the spatial patterns and progression of emergence within a multi-variable framework. We therefore introduce a new metric, the relative ToE ($ToE_{rel}$), to better compare ToE across different models and variables. $ToE_{rel}$ is computed by subtracting the global mean ToE from the ToE field. Absolute years of emergence are thus not considered by this metric and it only illustrates whether a signal emerges relatively early or late for a given model. We investigated whether anthropogenic T or $O_2$ changes emerge first and link patterns of ToE(T)-ToE($O_2$) to changes in apparent oxygen utilisation ($\Delta[$-AOU$]$) and ventilation of the thermocline. In addition, we also identified the processes for earlier/later detection in $O_2$ changes compared to temperature changes.

This multi-model study relies only on results from only four different model families (GFDL-ESM, HadGEM2-CC, IPSL, MPI-ESM and CESM), applied in nine model configurations. All model configurations available from CMIP5 that provide 3-dimensional fields for $O_2$ and T for the control, historical and future-RCP8.5 scenario simulations have been incorporated into the analysis. Nevertheless, using a larger model ensemble would increase confidence in our results (Knutti and Sedláček, 2013).

A limitation of our study is that all the Earth system models included have a relatively coarse resolution for simulating the complex processes in the $O_2$ minimum zones (Margolskee et al., 2019). Earth System models diverge in projecting physical and biogeochemical changes in these regions (Brandt et al. 2015; Cabré et al. 2015). Some models used in this study project a large increase in [-AOU] (Fig. 8) and considerable warming (Fig. S6) in the eastern tropical Atlantic, likely indicative of a reduced upwelling (Gnanadesikan et al., 2007). Observations show a decrease in $O_2$ and an expansion of hypoxia in the tropics (Stramma et al., 2008, 2012) over recent decades, contradicting the long-term projections from some models. However, these observed trends in the tropics may also be a result of natural variability acting on multi-decadal timescales associated with the Pacific Decadal Oscillation.

Comparing ToE estimates from different studies is delicate due to the model and method dependencies of ToE. Although the generic definition of ToE is under consensus, the methodologies applied to estimate ToE differ in the published literature as mentioned in the introduction (e.g. IPCC (2019)). Depending on the spatial and temporal scale of a given variable, the threshold for which emergence is defined and the reference period applied, the absolute value of ToE can differ. In addition, the ToE also depends on the definition of the background variability, here acting as noise (Hameau et al., 2019). Estimating the background noise as the standard deviation (SD) of the internal chaotic variability from the control simulation (Frölicher

et al., 2016), or as the SD of the variability from the industrial period (after removing anthropogenic trends; Keller et al., 2015; Henson et al., 2016) result in earlier ToE for both $O_2$ and T as when estimating the noise from the total (internal and externally-forced) natural variability over the last millennium. Yet, the finding that anthropogenic $O_2$ change emerges before anthropogenic warming in large ocean regions is robust across investigated choices. The anthropogenic signal is frequently

computed as a linear trend over a few decades (Rodgers et al., 2015; Henson et al., 2017; Tjiputra et al., 2018). However, the resulting slope depends on the time window used to calculate the linear trend. Hameau et al. (2019) use a low-pass filtered output to estimate the signal. They showed that ideally the noise ($N$) component of ToE should be estimated from simulations that include natural variability forced by explosive volcanic eruptions and changes in total solar irradiance, especially when assessing regional to global scale ToE estimates. However, these authors also find that on a grid cell scale, internal variability

is typically the dominant contribution to overall natural variability during the last millennium. Therefore, estimating the noise from control simulations that include internal variability only, as done in this study, appears justified.

Another limitation of our study lies in the assumption that the anthropogenic signal emerges from interannual to multi-decadal internal variability. The anthropogenic signal $S$ and the noise $N$ is estimated by smoothing the model output with a

multi-decadal spline filter. Any potential natural centennial variations are retained in the signal $S$ and removed from the noise $N$. Results from a forced simulation over the past millennium with CESM1.0 show that potential biases in ToE arising from the neglect of long-term natural variability are small for this model (Hameau et al., 2019). However, our multi-model analysis reveals centennial variations in some grid cells and models causing multiple emergence of the signal from the noise (Fig. A2). This may bias the detection of the anthropogenic signal towards early emergence. Here, we constrained detection to partly

circumvent problems with re-emerging signals; we require that the trend of the signal at the time of emergence must have the same sign as the change between the last and first 30 model years. Re-emerging signals are found in only a few grid cells, except in HadGEM2, and centennial natural variability appears to play a minor role in these simulations. We expect therefore that our estimates of ToE are reliable for the model ensemble.

Published studies addressing the detection of anthropogenic ocean warming focus on temperature at sea surface. To our knowledge, only a single study Hameau et al. (2019) using output from a single model is assessing ToE(T) in the thermocline. Yet, the thermocline is habitat for many fish and other species. Warming in combination with other stressors, such as deoxygenation, ocean acidification and hypocapnia, may reduce the habitat suitability of marine ecosystems in a future climate (e.g. Deutsch et al., 2015; Gattuso et al., 2015; Breitburg et al., 2018; Cheung et al., 2018). Multi-tracer analyses contribute to a

better understanding of the potential impact on marine ecosystems in a changing ocean.

We find that thermocline anthropogenic warming emerges first in low latitudes, followed by the Southern Ocean and the high northern latitudes. No emergence is detected in parts of the subtropical gyres of the Pacific and Indian Ocean. The rapid emergence at low latitudes is explained by the small internal variability, but moderate to strong warming signals. Exceptions

are the subtropical gyres in the Atlantic, where it takes approximately two additional decades to detect the temperature changes,

mainly because of the relatively large internal variability there. The warming in mid- to high latitude thermocline emerges approximately 60 to 80 years later than in low latitudes. No emergence is simulated for the Pacific and Indian subtropical gyres, because the changes in temperature are relatively small and the internal variability relatively high there (in accordance with Hameau et al., 2019). For comparison, surface temperature changes emerge at first in low latitudes and then in midlatitudes
(Henson et al., 2017).

The time of emergence spatial pattern of thermocline oxygen changes is almost opposite to the one of temperature. Rapid emergence for $O_2$ is simulated at midlatitudes, whereas low latitudes generally do not experience emergence of the $O_2$ signal by the end of the 21st century (Rodgers et al., 2015; Frölicher et al., 2016; Long et al., 2016). Although internal variability
is low in the tropical regions, the $O_2$ signal does not emerge by 2100. This is because the projected changes are also small. This is due to the opposite responses of $O_2$ components. The thermal component is simulated to decrease (due to temperature increase), but [-AOU] is on average projected to increase, counteracting the $O_{2,\text{sol}}$ trend (Frölicher et al., 2009; Cocco et al., 2013; Bopp et al., 2017). Some regions show similar relative ToE but for different reasons. For example, in the North Pacific subtropical gyre and the Southern Ocean, both the oxygen depletion and the natural variability are relatively strong. In the
Arabian Sea, internal variability and anthropogenic response are both rather weak. Nevertheless, the $S/N$ ratio results in very similar relative ToE for all these regions.

The transient climate response of the individual models and therefore the ocean heat uptake, thermocline warming and deoxygenation can substantially differ among models (Bopp et al., 2013). The simulated internal variability also differs con-
siderably across models (e.g. Resplandy et al., 2015; Frölicher et al., 2016). ToEs computed from CESM1.0 projections, for example, differ by many decades in absolute values from other CMIP5 models, mostly due to a very weak internal variability. Nijsse et al. (2019) suggest that the magnitude of simulated decadal variability and climate sensitivity might be correlated. They suggest that models with a high climate sensitivity tend to simulate a high decadal variability. This may imply a compensation between the simulated signal and noise on the decadal scale. To extract valuable insights as to the relative spatial and
temporal features of emergence across models and variables, we introduced a new metric, the relative time of emergence. By normalising the ToE using the globally averaged ToE as reference allows for a more direct comparison with the other models. As a result, the patterns and time of emergence of anthropogenic changes in $O_2$ and warming in CESM1.0 are more coherent with the other models for ToE$_{\text{rel}}$ than for the absolute ToE.

Following Hameau et al. (2019), we compared the ToE(T) with the ToE($O_2$) in nine models. We find that the anthropogenic decline in $O_2$ emerges before anthropogenic warming in a significant part of the thermocline. On average across the nine models, an area covering 35±11 % of the global thermocline shows emergence in $O_2$ change before temperature change. Yet, the exact locations of these patterns differ across models. Only 17 % of the global thermocline show agreement (seven out of the nine models) on earlier emergence of $O_2$ changes prior to T changes, Thus, our multi-model analysis confirms earlier
findings using output from a single model only (Hameau et al., 2019). The early emergence of $O_2$ suggests that the monitoring

of biogeochemical variables would be particularly useful to detect early signals of anthropogenic change in the ocean interior (Joos et al., 2003). Multi-tracer observations of both physical and biogeochemical variables may enable an earlier detection of potential changes than temperature-only data (Keller et al., 2015) in specific regions and for specific processes.

5      Hameau et al. (2019) established a direct link between the early emergence in $O_2$ with a slow down of ventilation. A weaker ventilation leads to a decrease in [-AOU], and therefore to a reduction in $O_2$, with a minor role for organic matter export changes in their simulation. We used [-AOU] as a ventilation age proxy for our model ensemble and concluded that the slow down of the ventilation induces $O_2$ changes to be detectable before T changes in many regions. A slower ventilation seems to shift the balance between $O_2$ supply from the surface and $O_2$ consumption by organic matter remineralisation. Moreover, a
10 more stratified upper ocean delays the propagation of the temperature signal from the surface into the subsurface waters. Note that the exact locations of early $O_2$ emergence and reductions in [-AOU] and ventilation diverge among the models. This is partly due to model biases in terms of ocean dynamics. In addition, the use of depth coordinates to define a thermocline layer from 200 – 600 m may lead in our analysis to the inclusion of different water masses for different models. Another approach would be to perform the analysis on isopycnal levels instead on depth levels.

     To conclude, normalising ToE across models (relative ToE) or estimating ToE in relation to another variable (ToE(T) - ToE($O_2$)), reduces the multi-model spread arising from method and model dependencies. We find that in about 35 % of the thermocline anthropogenic $O_2$ depletion emerges before anthropogenic warming. This relative early emergence of $O_2$ is linked to a more sluggish ventilation of these subsurface waters under global warming. Our study also suggests that temperatures in
20 the thermocline have already left the bounds of internal variability in much of the tropical ocean and that temperatures will have left these bounds in most of the thermocline by 2100 under unabated global warming.

*Data availability.* The CMIP5 simulations are available on https://esgf-node.ipsl.upmc.fr. The CESM1.0 simulations are available upon request.

## 5   Figures

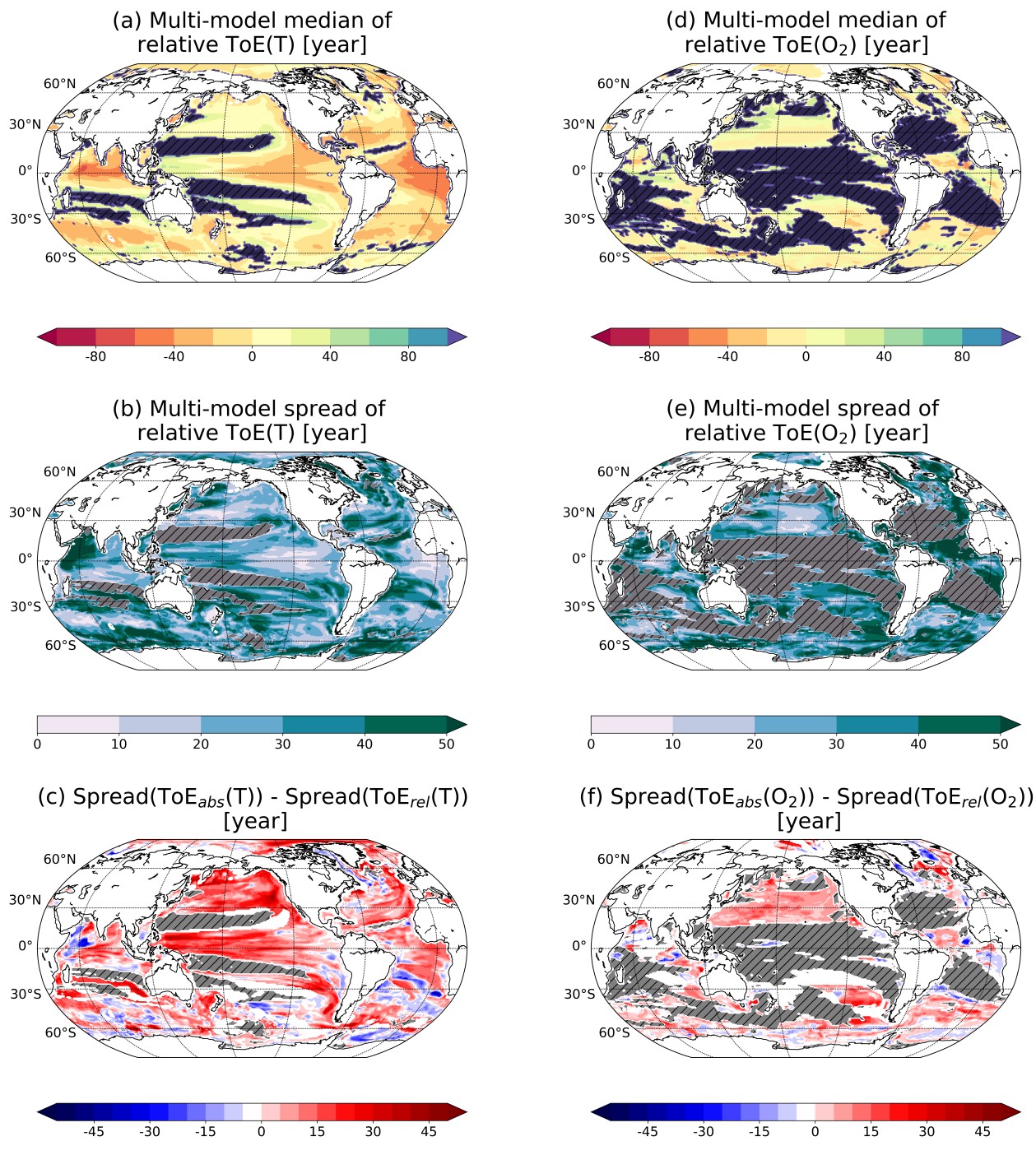

**Figure 1.** Multi-model median (top panel) and spread (middle panel) of relative ToE for temperature (left column) and dissolved oxygen (right column) for the thermocline (200 – 600 m). The spread is computed as the interquartile range. Difference ( lower panel)between the multi-model spread of absolute ToE estimates with the multi-model spread of relative ToE estimates for (c) temperature and (f) dissolved oxygen. The hatched areas show regions with no emergence for at least 3 models. For temperature (oxygen), the relative ToE estimates are shown for each model in Fig. 2 (3) and the absolute estimates in Fig. S1 (S2).

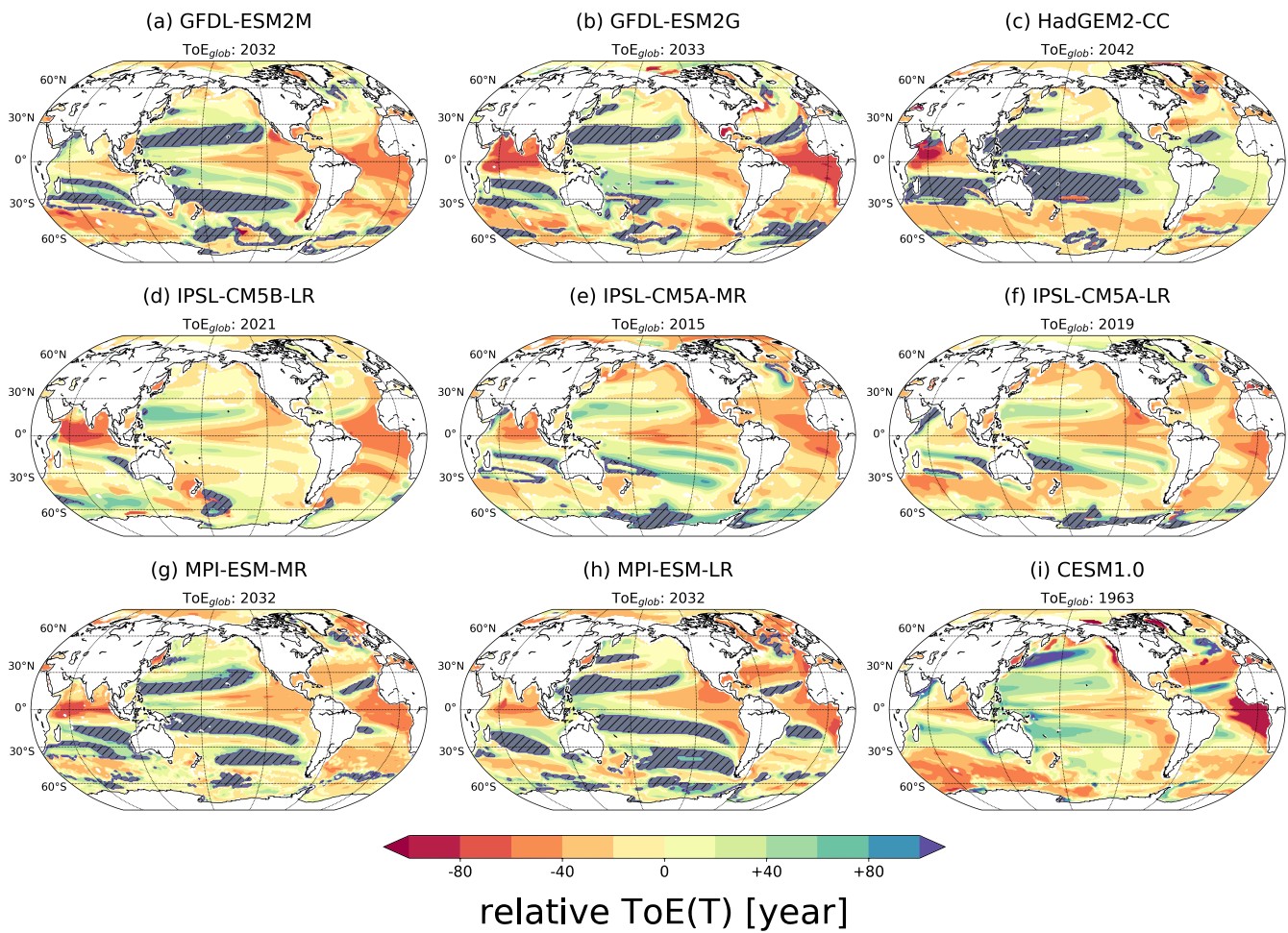

**Figure 2.** Time of Emergence (ToE) of T in the thermocline (200 – 600 m) relative to the averaged ToE in that layer for each simulation. The hatched areas show regions with no emergence by the end of the 21st century. The values of the global average ToE, ToE$_{glob}$, are given above each panel. The absolute ToE estimates are shown in Fig. S1.

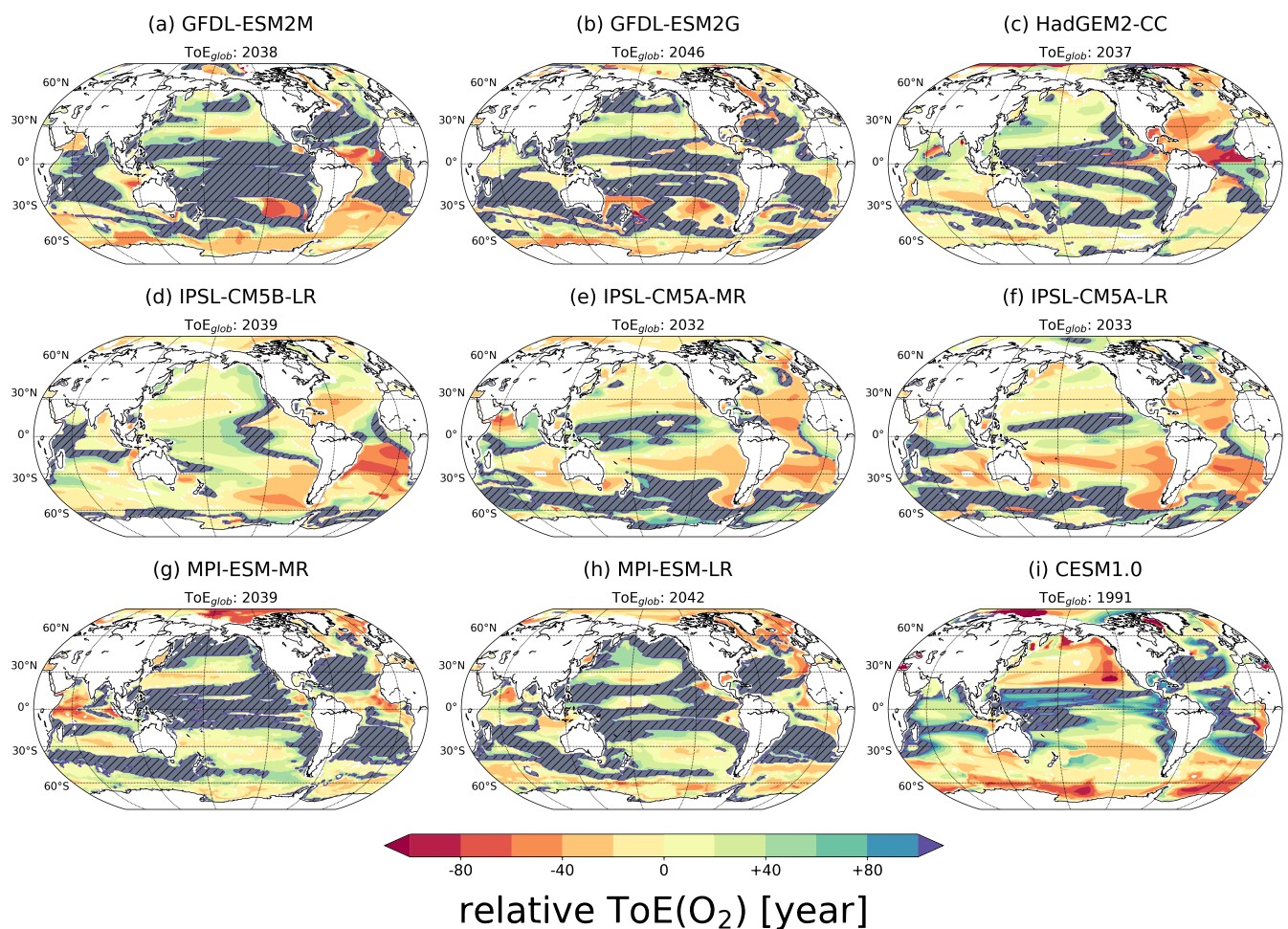

**Figure 3.** Time of Emergence (ToE) of $O_2$ in the thermocline (200 – 600 m) relative to the averaged ToE in that layer for each simulation. The hatched areas show regions with no emergence by the end of the 21st century. The absolute ToE estimates are shown in Fig. S2. The global average ToE, $ToE_{glob}$, is shown for each model

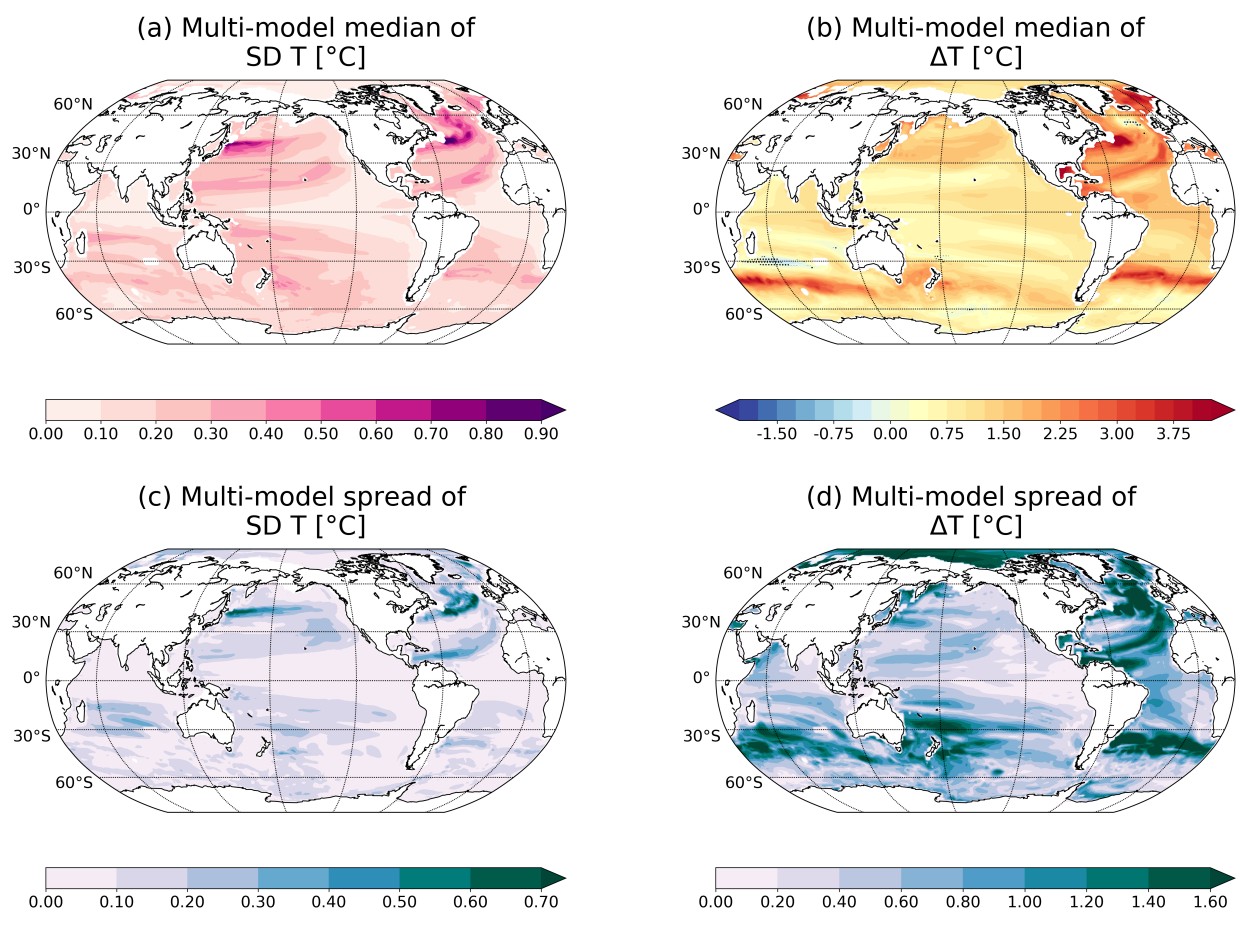

**Figure 4.** Median (top panels) and spread (bottom panels) of multi-model natural variability (standard deviation of control simulation; left panels) and changes by the end of the 21st century (right panels) of ocean temperature between 200 and 600 m. The individual responses for each model are shown in Figs. S3 and S4.

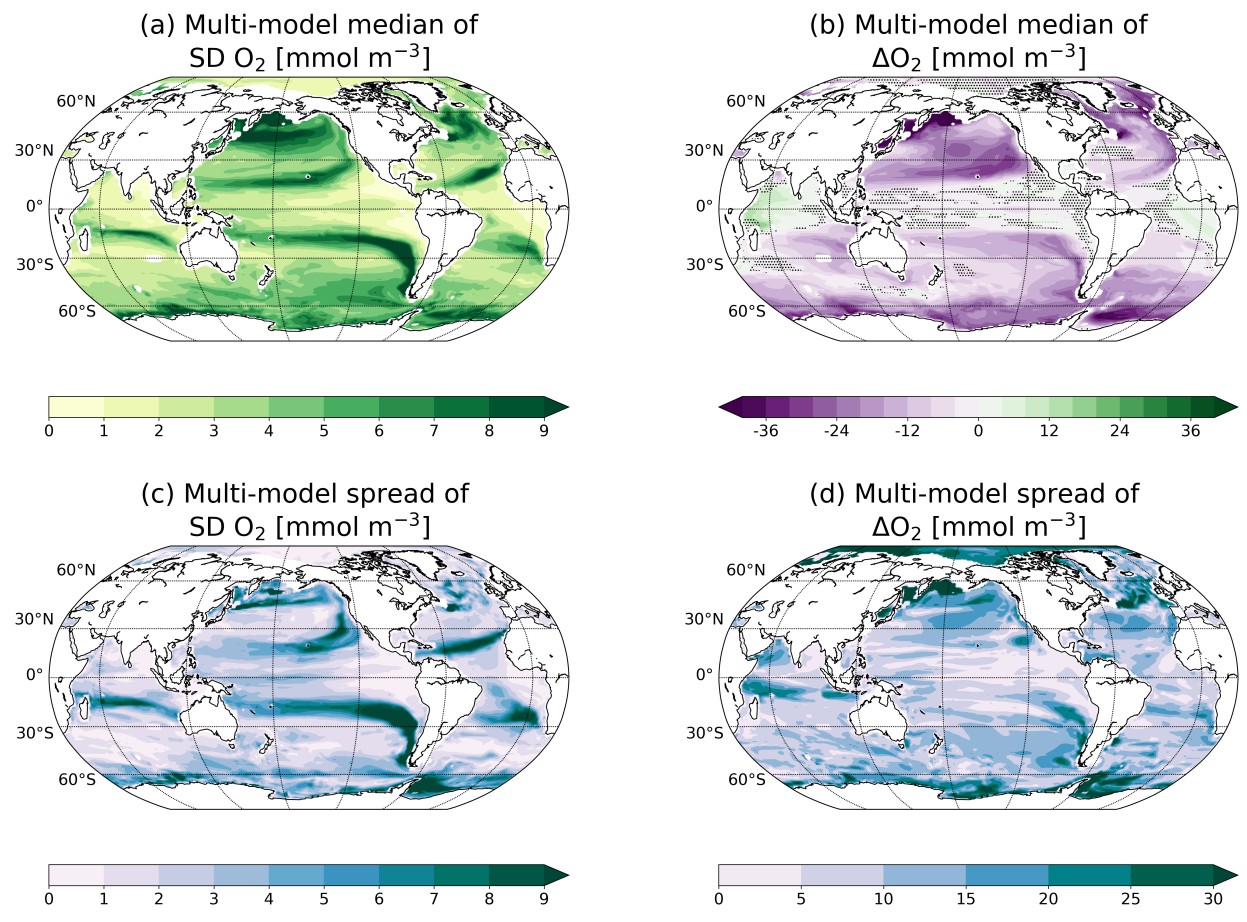

**Figure 5.** Median (top panels) and spread (bottom panels) of multi-model natural variability (standard deviation of control simulations; left panels) and changes by the end of the 21st century (right panels) of $O_2$ between 200 and 600 m. The hatched areas in panel b show regions where at least 70 % of the models do not agree on $\Delta O_2$ sign. The individual responses for each model are shown in Figs. S5 and S6.

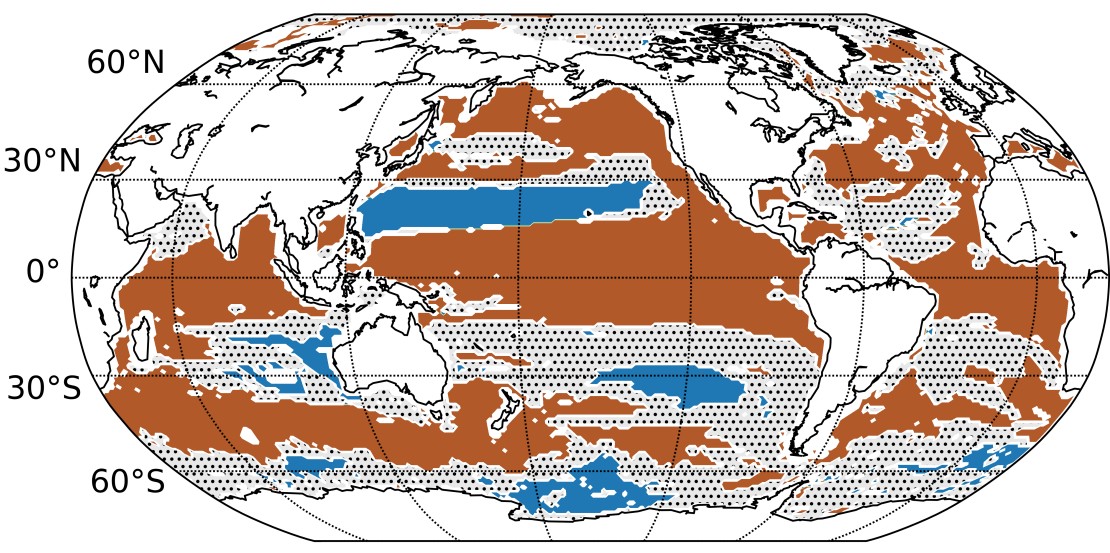

**Figure 6.** Summary map showing the regions where oxygen changes emerge before temperature changes (blue areas; ΔToE>0) and where temperature changes emerge before oxygen changes (brown areas; ΔToE<0) for at least seven out of nine models. The dashed areas show the regions where more than three models differ in the sign of ΔToE.

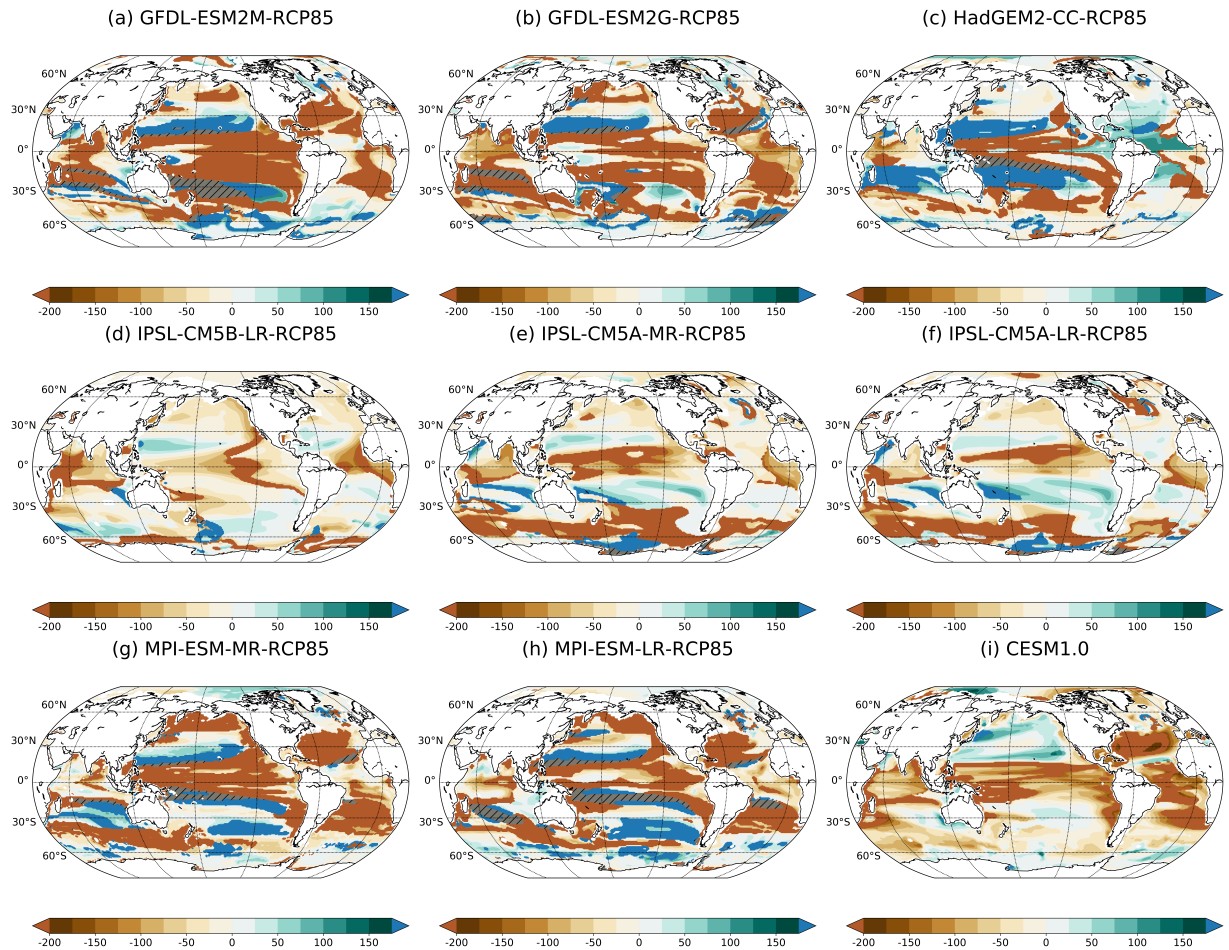

**Figure 7.** ToE(T) minus ToE(O$_2$) for each simulation in the thermocline. Blueish colours indicate earlier emergence of oxygen. Brownish colours indicate earlier emergence of temperature. The saturated colours mean that one of the variables has not emerged by 2099. No emergence in both T and O$_2$ are shown by the hatched areas.

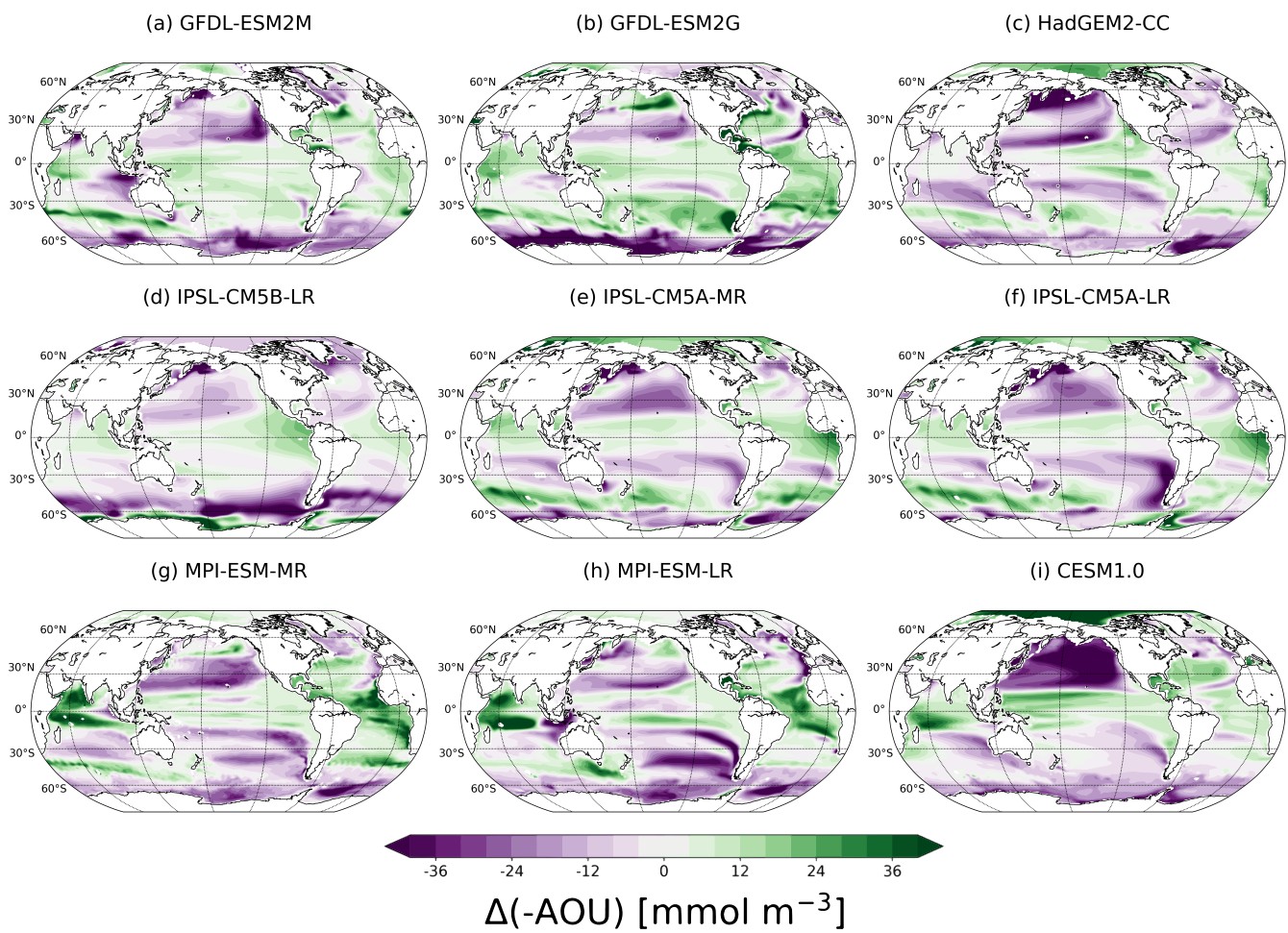

**Figure 8.** Anthropogenic changes ((2070-2099 CE) minus (1861-1959 CE)) in [-AOU] in the thermocline for each model.

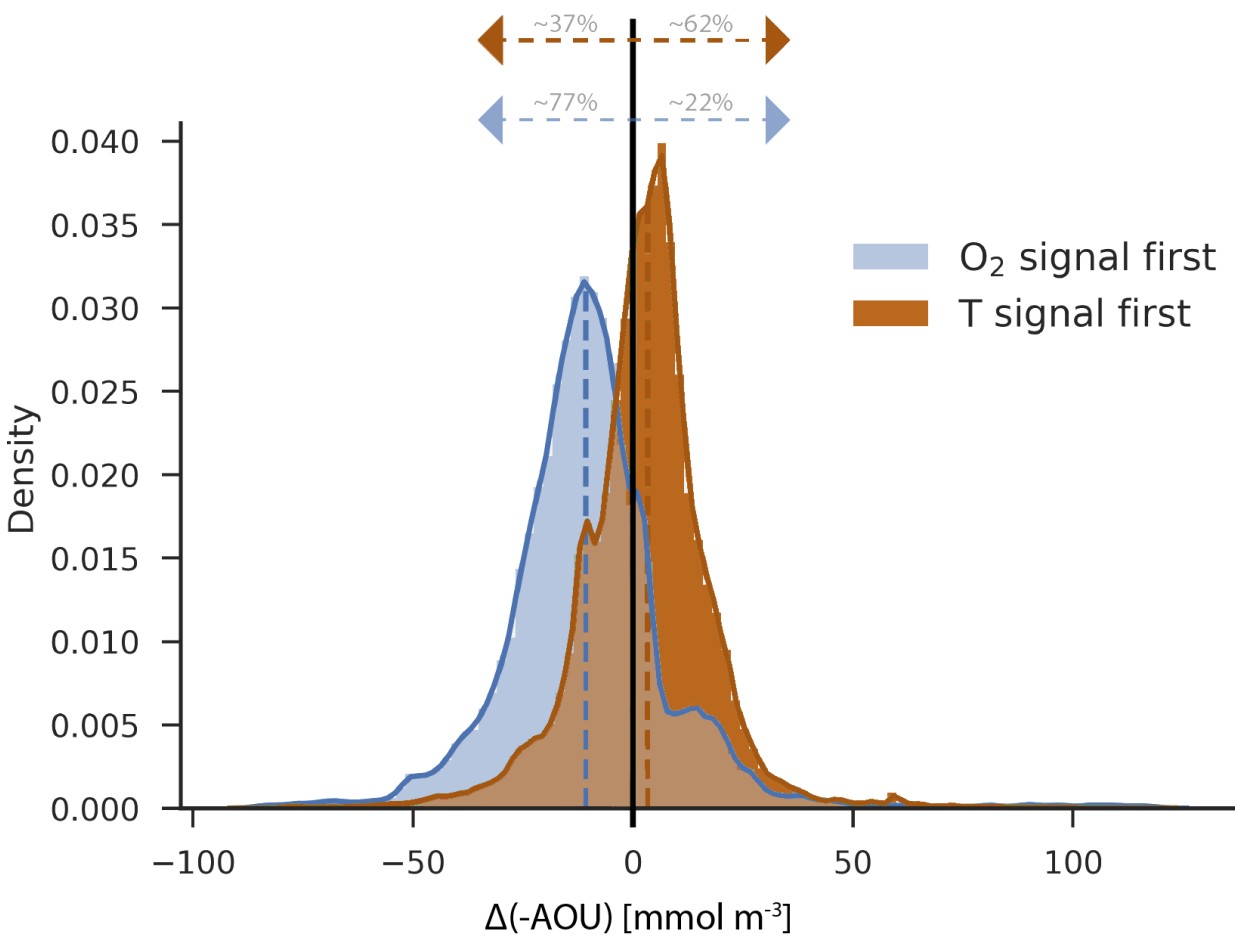

**Figure 9.** Density distribution of [-AOU] changes by 2099 for the grid points where the $O_2$ signal emerges first (blue) and where the temperature signal emerges first (brown) in the thermocline for the ensemble of 9 models. Each distribution is centred around the median (dashed blue: -10.8 mmol m$^{-3}$; dashed brown: 3.3 mmol m$^{-3}$).

**Table 1.** Overview of the Earth system models used in this study, their configurations and vertical and approximated horizontal resolutions.

| Earth system model | Physical ocean model | Biogeochemical ocean model | Vertical and horizontal ocean resolution |
|---|---|---|---|
| CESM1.0<br>Hurrell et al. (2013) | POP2 (Smith et al., 2010; Danabasoglu et al., 2011) | BEC (Moore et al., 2002, 2004) | 60 levels<br>~1° x 1° |
| GFDL-ESM2M<br>GFDL-ESM2G<br>Dunne et al. (2012, 2013) | MOM4p1 (Griffies et al., 2011)<br>GOLD (Hallberg, 1997) | TOPAZ2 (Dunne et al., 2013) | 50 levels<br>~1° x 1° |
| HadGEM2-CC<br>Collins et al. (2011) | HadGEM2 (Collins et al., 2011) | HadOCC (Palmer and Totterdell, 2001) | 40 levels<br>~2° x 2° |
| IPSL-CM5A-LR<br>IPSL-CM5A-MR<br>IPSL-CM5B-LR<br>Dufresne et al. (2013) | OPA (Madec et al., 2017) | PISCES (Aumont and Bopp, 2006) | 31 levels<br>~2° x 2° |
| MPI-ESM-LR<br><br>MPI-ESM-MR<br><br>Giorgetta et al. (2013) | MPIOM (Jungclaus et al., 2013) | HAMOCC5.2 (Ilyina et al., 2013) | 40 levels<br>~1.5° x 1.5°<br>40 levels<br>~0.4° x 0.4° |

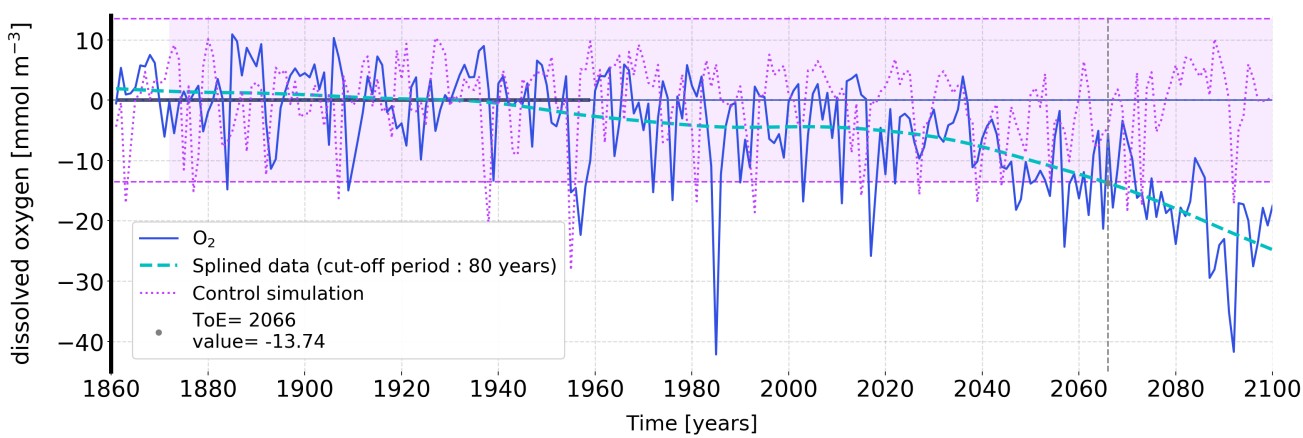

**Figure A1.** Illustration of the ToE method by using simulated oxygen concentration of the MPI-ESM-LR model averaged over 200 – 600 m in one grid cell in the Western North Pacific. The dark blue line shows that temporal evolution of oxygen anomalies from 1860 to 2099 relative to pre-industrial concentrations (1860 – 1959). The two pink lines indicate the magnitude of the noise, $N$, which is computed as two standard deviations from annual output of the control simulation. The anthropogenic signal, $S$, is the simulated oxygen concentration over 1860 to 2090 from the forced simulation splined with a 80 year cut-off period (dashed cyan). Here, the resulting ToE is the intersection between the lower limit of the noise and and the splined signal (vertical dashed line).

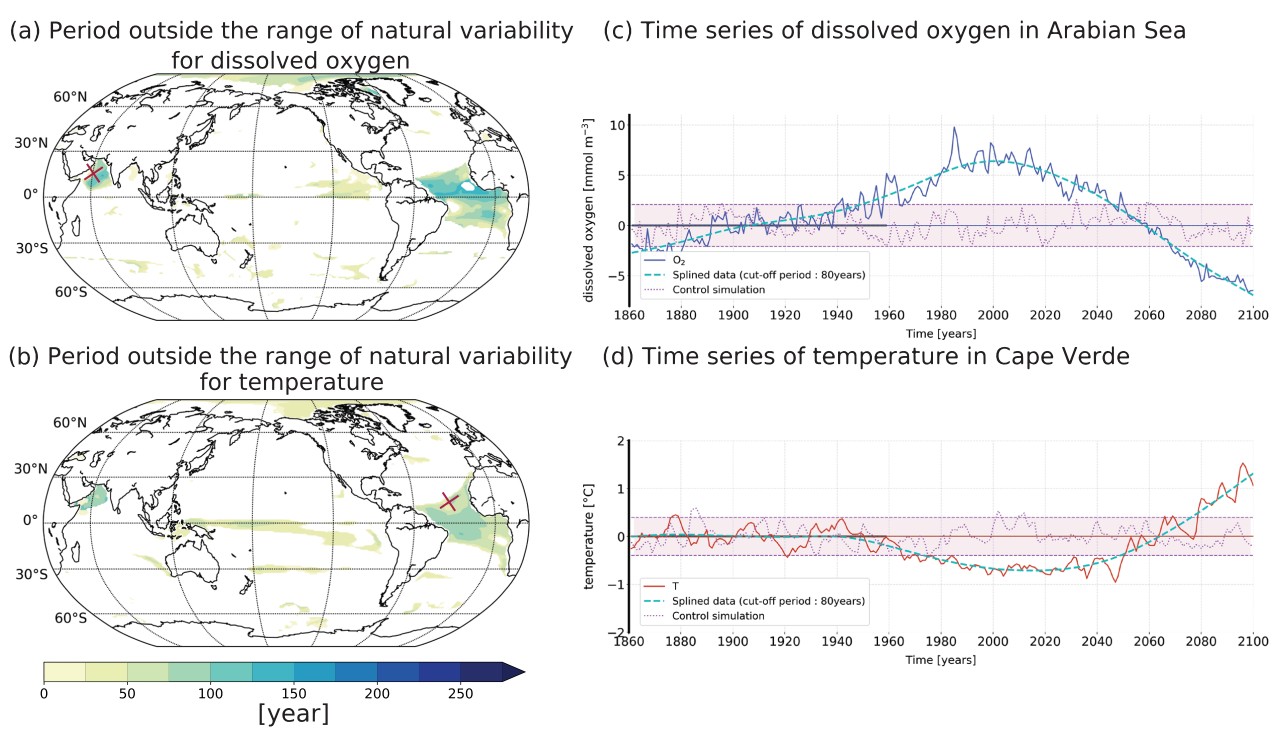

**Figure A2.** Period outside the range of natural variability for (a) oxygen concentration and (b) temperature in the thermocline for the model HadGEM2. The time series show two example of the temporal evolution of the oxygen concentration (c) and temperature (d) for a single grid point (red crosses in the left panels: Arabian Sea (c) and equatorial Atlantic (d)).

*Author contributions.* All authors contributed to the discussion and the writing of the paper.

*Competing interests.* The authors declare that they have no conflict of interest.

*Acknowledgements.* A. Hameau, F. Joos and T. Frölicher thank the Oeschger Center for Climate Change Research, the Swiss National Science Foundation (#200020_172476 and PP00P2_ 170687), and the European Union for financial support and the CSCS Swiss National Supercomputing Center for computing resources. This publication has received funding from the European Union's Horizon 2020 research and innovation programme under grant agreement No 820989 (project COMFORT, Our common future ocean in the Earth system – quantifying coupled cycles of carbon, oxygen, and nutrients for determining and achieving safe operating spaces with respect to tipping points). The work reflects only the authors' view; the European Commission and their executive agency are not responsible for any use that may be made of the information the work contains. We thank C. Raible and F. Lehner for providing CESM output. We also thank the World Climate Research Programme's Working Group on coupled Modelling, which is responsible for CMIP5, and the climate modelling groups for producing and making available their model output. We thank the two anonymous referees and Sarah Schlunegger for their effort and time and their comments.

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

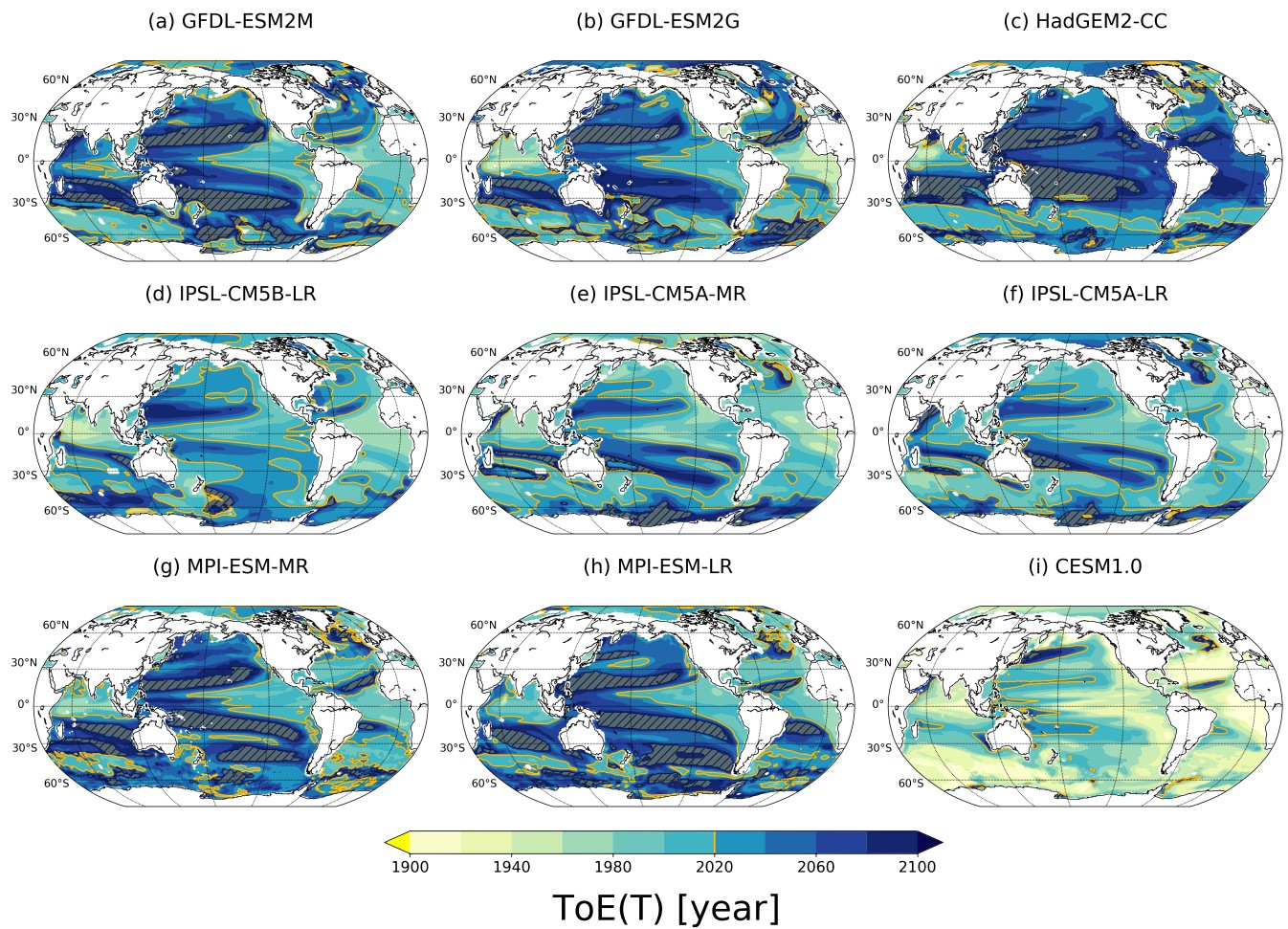

**Figure S1.** Time of Emergence (ToE) of T in the thermocline (200 – 600 m). The hatches areas represent the regions where the signal has not emerged by the end of the 21st century. The yellow contours highlight the present time (year: 2020).

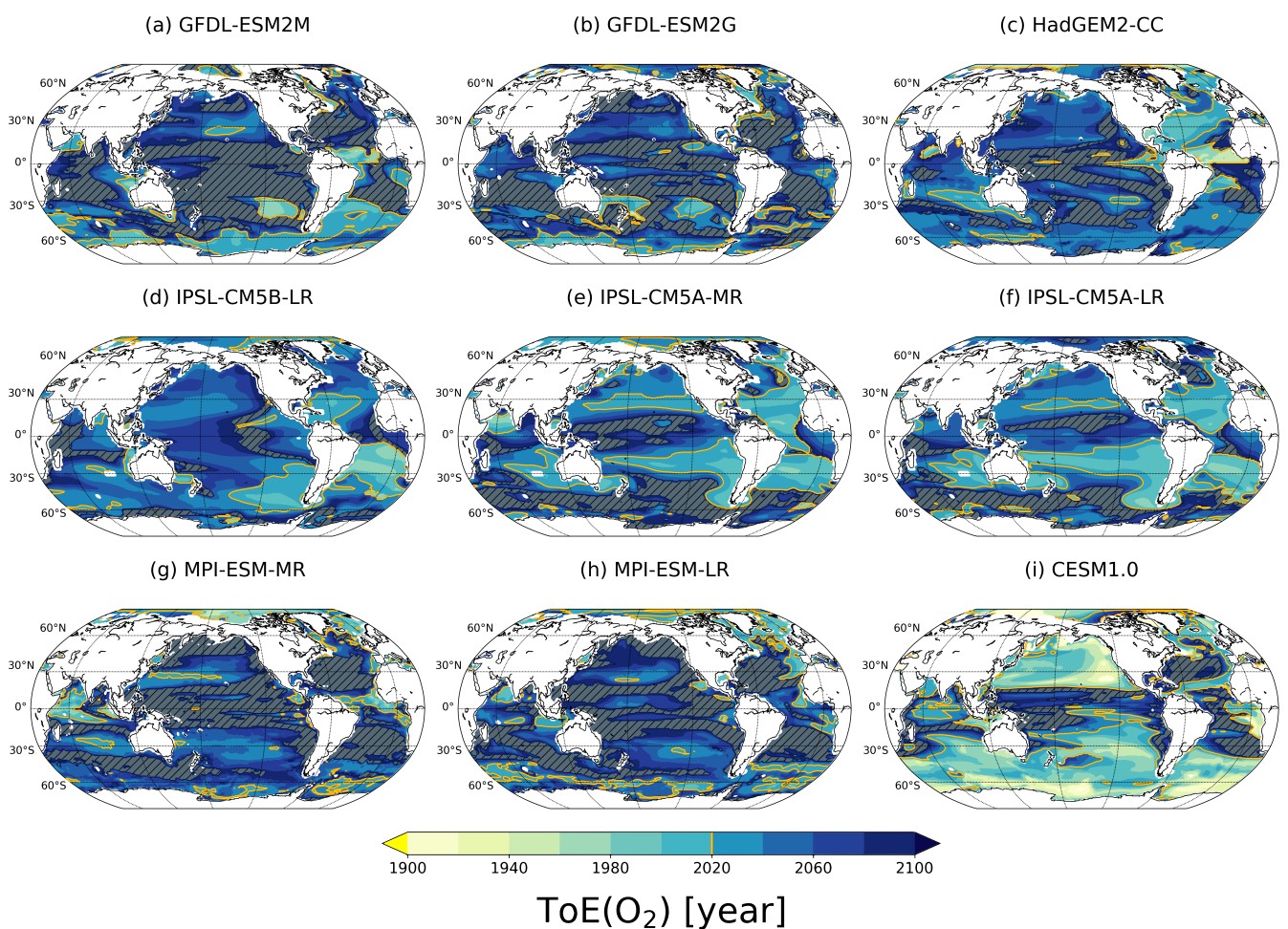

**Figure S2.** Time of Emergence (ToE) of $O_2$ in the thermocline (200 – 600 m). The hatches areas represent the regions where the signal has not emerged by the end of the 21st century. The yellow contours highlight the present time (year: 2020)

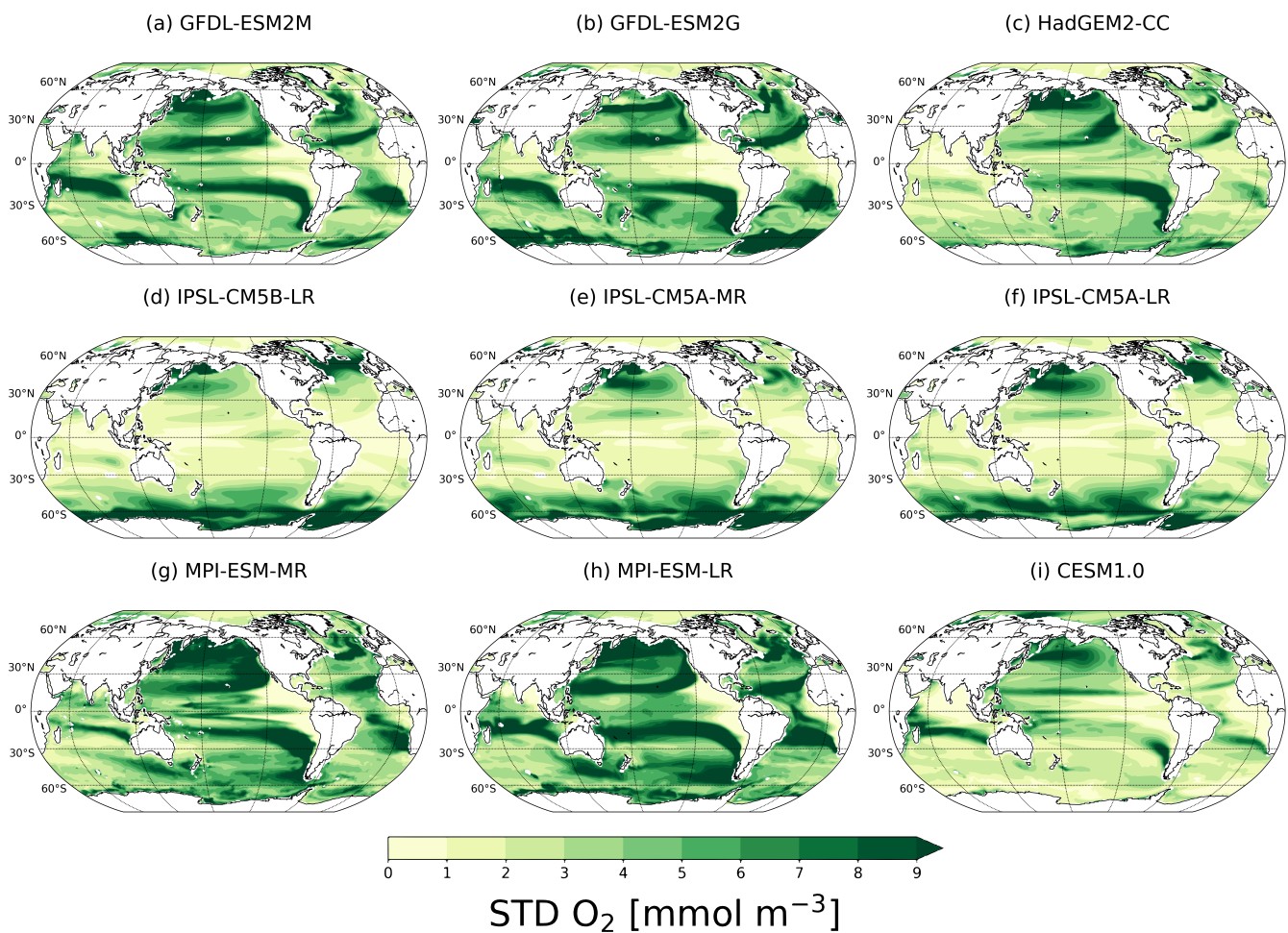

**Figure S3.** Standard deviation of temperature in the averaged layer 200 – 600 m of the associated control simulation.

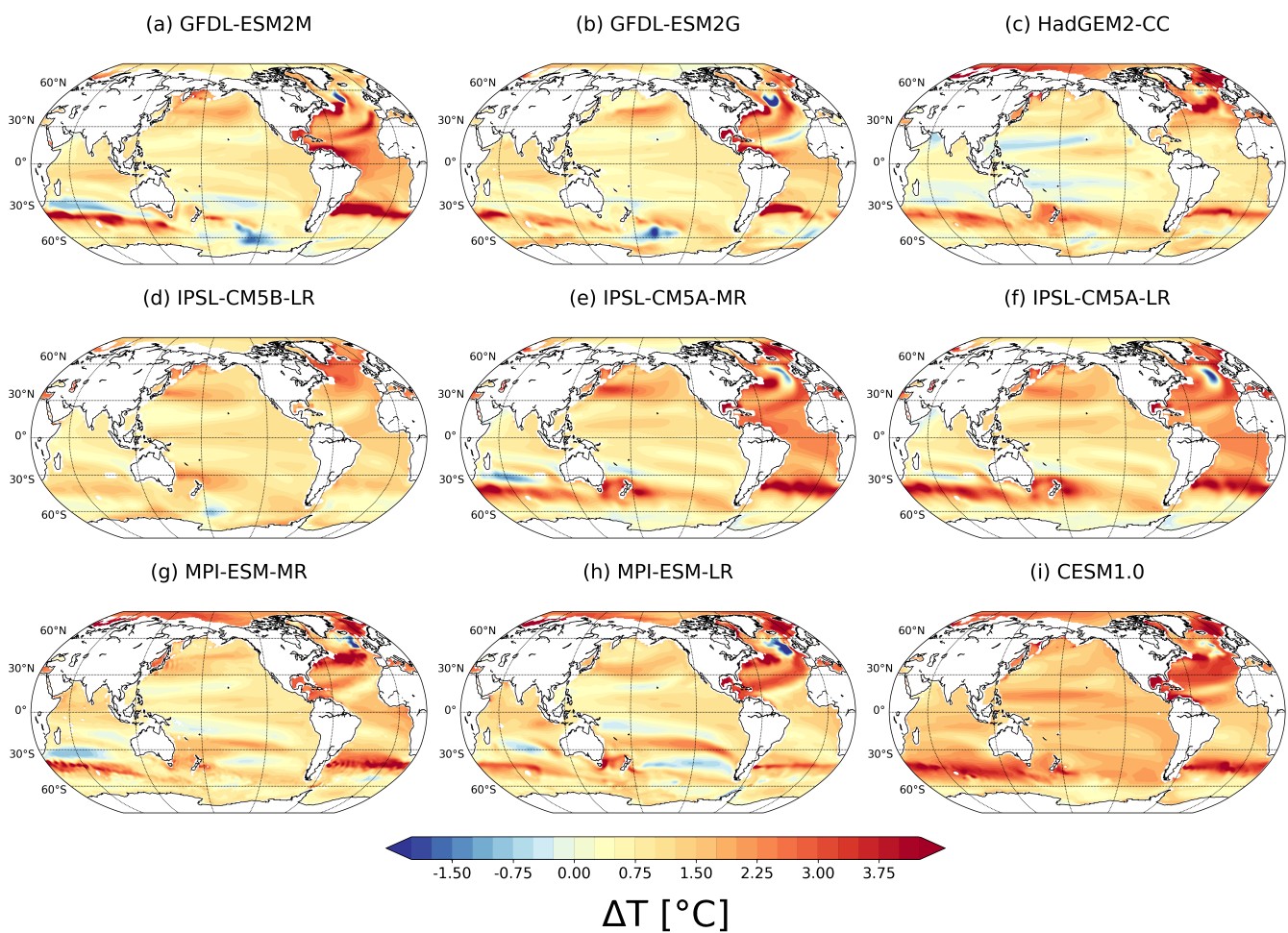

**Figure S4.** Anthropogenic changes ((2070-2099 CE) minus (1861-1959 CE)) in temperature.

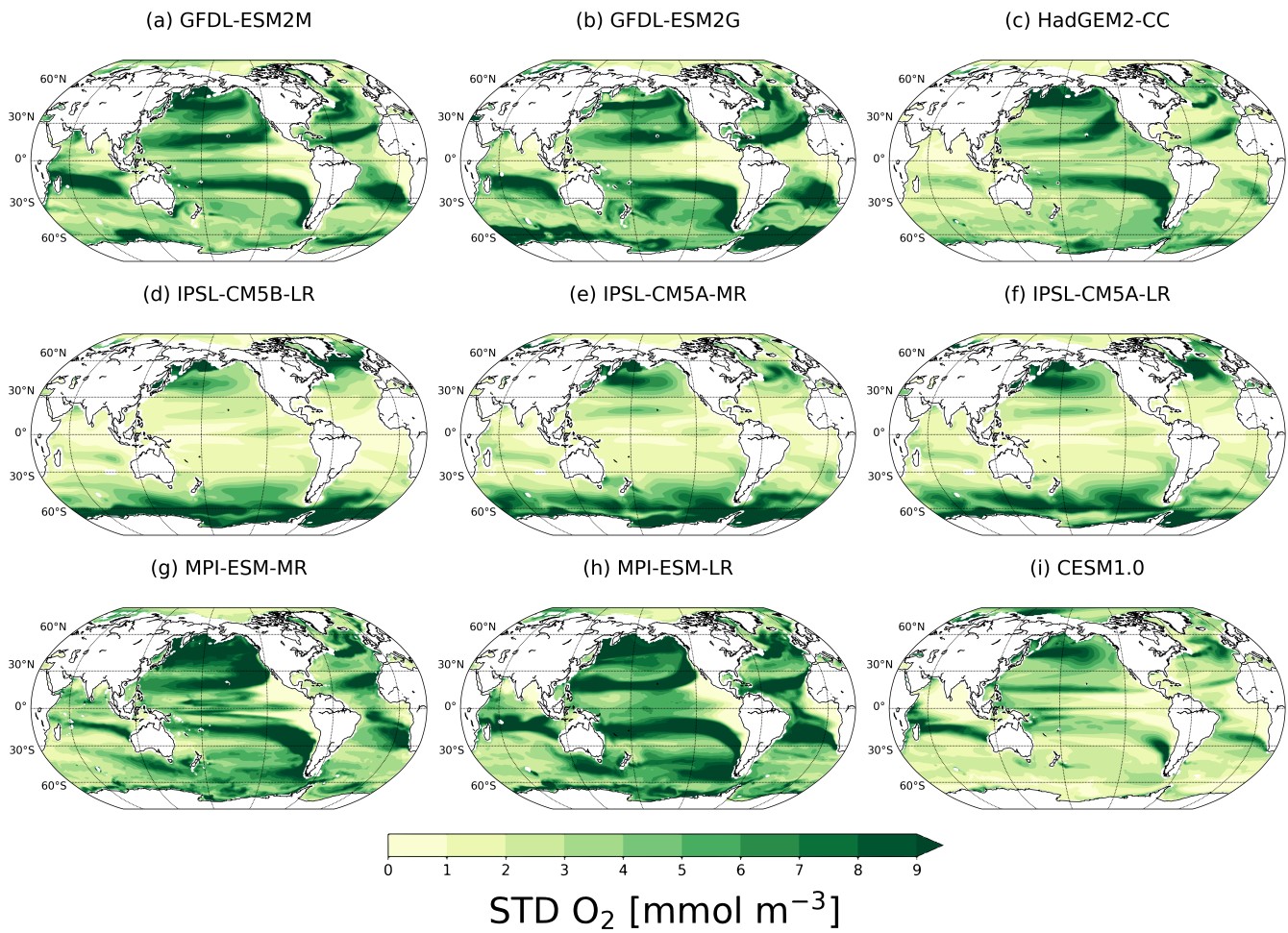

**Figure S5.** Standard deviation of $O_2$ in the averaged layer $200 - 600$ m of the associated control simulation.

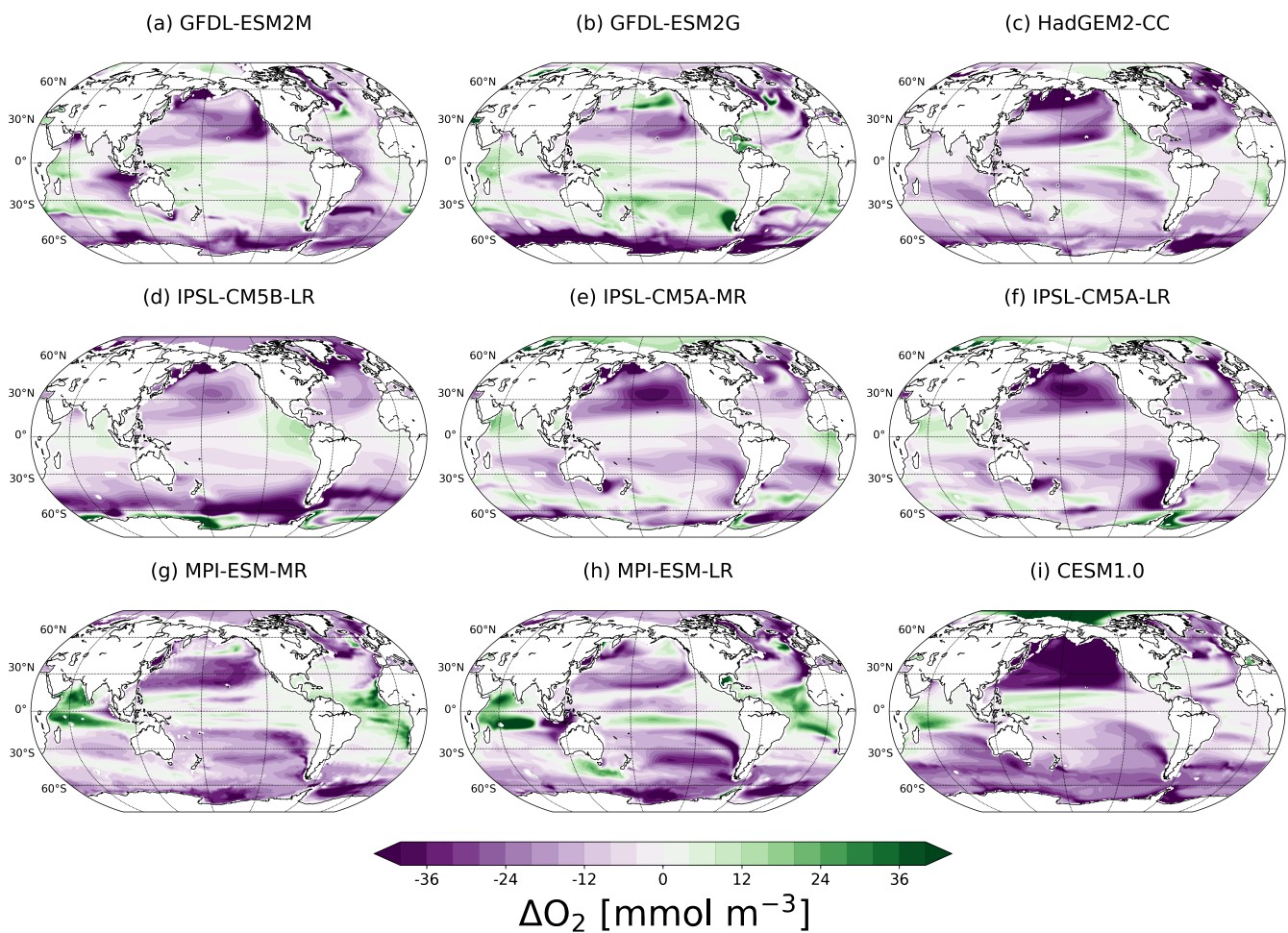

**Figure S6.** Anthropogenic changes ((2070-2099 CE) minus (1861-1959 CE)) in $O_2$.