# Peer review of "Is deoxygenation detectable before warming in the thermocline?"

_Biogeosciences, 2019_

## Referee Comment (RC1) · Sarah Schlunegger (Referee) · 17 Oct 2019

Sarah Schlunegger (Referee)

ss23@princeton.edu

General:

Hameau et. al. provides a multi-model (CMIP5) assessment of the relative timing of emergence of anthropogenic change in thermocline oxygen and thermocline temperature. To facilitate multi-model assessment of time of emergence (ToE), they provide a new metric, relative ToE. They find that for most of the global ocean changes in temperature emerge prior to changes in oxygen, however for some regions, changes in oxygen emerge prior to changes in temperature. Anthropogenic changes in oxygen emerge prior to temperature in regions where reduced solubility and ventilation work in tandem to reduce oxygen concentrations. In these locations, reduced ventilation also

slows the propagation of anthropogenic warming signals from the upper ocean into the ocean interior, further contributing to the delay between emergence of thermocline oxygen and temperature.

This paper represents an important contribution to the growing body of mechanistic interpretations of emergence timescales in the global ocean. Additionally, it confronts the known challenge of model-intercomparison with provision of a simple, yet powerful new metric, relative ToE. I recommend its publication in Biogeosciences, but only after minor-to-moderate revision of the text and inclusion of an additional figure to better visualize the regions for which thermocline oxygen robustly emerges prior to temperature.

Sincerely, Sarah Schlunegger, PhD Princeton University Program in Atmospheric and Oceanic Science

Specific:

1. Improve Abstract In the abstract, I would suggest the following rewording to replace some of lines 7-10: Changes in thermocline oxygen emerge prior to changes in temperature because oxygen declines occur due to the confluence (or additivity) of both reduced solubility and ventilation. Otherwise, the abstract could be further streamlined and focused. Feel free to borrow from my summary above.

2. Improve motivation/framing of the introduction Page 3 frames the question of "Which emerges first, physical or biogeochemical variables?" However, this is an ill-formed question, as the chronology of emergence has already been documented to not follow strict temporal separation of physical and biogeochemical variables. For example, Rodgers et al., 2015 evaluates ESM projection of 4 variables and find the following general emergence sequence: pH, SST, O2, NPP – biogeochemical, physical, physical, biogeochemical. Schlunegger et al., 2019, evaluates ESM projection ~20 physical/biogeochemical variables, finding their emergence timescales are separated by their association to either chemical (gas-exchange) or physical (warming) impacts

of climate change on the ocean, and NOT to whether the variable itself is physical or biogeochemical. Variables that are first-order impacted by increasing atmospheric pCO2 and increased gas-exchange, like DIC, pH etc., will emerge the most rapidly (carbonate-chemistry related biogeochemical). Later, as atmospheric warming propagates into the ocean (physical) warming signals emerge, and quickly after solubility induces changes (like O2sat). Even later, circulation eventually adjust (physical), thereby altering nutrient supply (biogeochemical), and subsequently primary production (biogeochemical), export production (biogeochemical), etc. The broad order of emergence could then be described as 1. Carbonate-chemistry related biogeochemical and biology (hard tissue pump), 2. Surface temperatures and solubility-related variables (like O2), 3. Ocean dynamics, 4. Biogeochemical impacted by ocean dynamics (nutrients, production).

3. Additional methodological explanation required Explanation of how S is computed, and its statistical properties relative to N, should be addressed more clearly. For instance, there are broadly two distinct usages of the term "time of emergence" – the first is to define the point in time at which the ocean state/variable is distinctly out of the range of the pre-industrial state, the second, is to define the point in time at which a forced trend is outside the range of how large natural trends are likely to be. I infer that the first meaning is used here, but that should be made clear.

4. Potentially reordering the figures to improve narrative flow Another immediately odd thing that I see is that Figure 1a and 1b at first glance oppose the main hypothesis of the manuscript. This is just visually the case because O2 has higher model disagreement and T and therefore more area where at least 2 of the models think emergence does not occur this century. However, I know many (mostly senior) scientists who just read the abstract, skim the method and the look at the figures. From this alone, the figures do not tell your story until Figure 6. In fact, Figure 1 is somewhat irrelevant to the stated hypothesis in the introduction. The order of figures could therefore be revised to Figure 2, 3, 6, 7, 8, 1,4,5. With the text also reordered.

[Figure]

5. Additional figure to better visualize regions of ToE(O) < ToE(T) These regions should be better visualized in the paper, potentially in a synthesis map showing locations of ToE(O)<ToE(T) robust across models (could use the 7/9 threshold for example) and/or a figure/bargraph for which regions are aggregated and the average relTOE(O) vs re-TOE(T) is ploted for each model. So the figure would have regions on the X axis and ToE(O)-ToE(T) on the Y axis. From there the reader could see the spread in the models as well as pick out regions where it is most likely for ToE(O) to be significantly shorter than ToE(T).

Additional Technical/Editorial Corrections:

Page 1: Line 3: remove word "as" Line 12-14: Simplify sentence with: To normalize across disparate trends and variability of the CMIP5 ensemble, we compute the local ToE relative to the global mean ToE within each model.

Page 2: Line 4-5: correct to remove "and"... adversely affect marine organisms, ecosystems, and the services they provide Line 6: remove "a": ...experienced significant warming Line 9: Make "scale" plural...on regional to local scales Line 13: Slightly awkward jump to discussion of ESMs. Line 17: Remove first phrase, "Concomitant with ocean warming" and just start with "Observation-based studies..." The mechanisms of observed oxygen decline is subsequently explained, so it needn't be partially explained initially. Lines 19-21: Break into 2 sentences and add context. "In subsurface waters, oxygen concentration is also affected by ventilation, remineralization of organic matter and air-sea disequilibrium. In the contemporary ocean, oxygen decreases are mostly dominated by a reduction in ventilation and increased consumption of oxygen during remineralization (references)." Line 21: change to... "The largest oxygen declines are located" Line 23: define "late" industrial period. Line 23-24: change to... "Therefore, it is challenging to distinguish human-caused trends from natural variations in the observational record of ocean O2." Line 24: remove "also"... Modelling studies agree on the sign of ... Line 24-26 could be combined instead with 13-16 and form their own paragraph discussing the current state of modeling O2 trends. Line 32: include reference,

" (e.g. Deutch et. al., 2015)"

Page 3 Line 11-12: rewrite. . ."is critical to understanding contemporary O2 and temperature changes." Line 17-29: rewrite "One study, (Hameau et al., 2019), uses a single model [ which model?], to investigate ToE of temperature in the thermocline, finding that anthropogenic ocean warming emerges much earlier than the O2 signal in low and midlatitude regions. Delayed emergence of changes in O2 is due to the opposing effects of decreases in O2 solubility and O2 consumption. In the high latitudes and the Pacific subtropical gyres, deoxygenation emerges before ocean warming in [ name model]. This occurs because decrease in oxygen solubility are reinforced by increased O2 consumption, leading to strong O2 depletion However, it is unknown if this single-model result is robust across a suite of different Earth system model simulations. Here, we conduct a multi-model study to more broadly test the hypothesis that anthropogenic deoxygenation in the thermocline emerges prior to anthropogenic warming. Since the primary objective is to test the consistency across models of the order of emergence (deoxygenation prior to warming) within a single model, we introduce and use a relative ToE to conduct the intercomparison, rather than the absolute year of ToE. We define relative ToE as. . . "

Page 5 Lines 1-10: Over what time frame is S estimated? The texts read "Enting, 1987"... Ending in 1987? At first I thought it was a typo but now I see it is a reference. To clarify, maybe state that S is a timeseires that extends from pre-industrial to 2100 (If I am understanding correctly). Secondly, with what "fitting?" –linear fit, polynomial, etc.? Perhaps the word "fit" should be excluded, if it is in fact meant to just say that the time-series is low-pass filtered. Line 20/21: change request to require and is to are: "We therefore require that ToE values are defined for at least seven out of nine models to compute the multi-model statistics (median and spread)."

Line 24: New section, entitled "Separating mechanisms of oxygen change" or something like this.

Line 24-25: Potentially rewrite: To diagnose processes driving the simulated changes in ocean O2, the direct thermal/solubility component of change (O2sat) can be isolated from the total O2 change. The residual, Apparent Oxygen Utilization (AOU), represents the summation of all non-thermal changes, including those resulting from changes in ventilation and remineralization.

Page 9 The early emergence of T in the Eastern Equatorial pacific should be discussed, and if possible, mechanistically explained. At first glance, it is surprising that an area of such interannual variability at the surface can have relatively little variability at intermediate depths. Line 20: remove word 'typically'

Page 10 Line 11: remove word 'surprisingly' Line 12: what is the confidence interval referenced? Is this references figure 1e, the multi-model spread? If the medium relTOE is 10 and the spread is also 10, then what does "confidence interval" mean in this context? And in that case, a confidence interval value of 1 is not "high". Usually confidence interval terminology is used to describe the probability of rejecting of a null hypothesis. The implicit hypothesis of the statement would be that in the CMIP5 ensemble, this region's relTOE(O2) does not significantly differ from the global average TOE(O2). Potentially a better way to convey this is to state that the multi-model ensemble agrees that the North Pacific represents a region for which emergence timescales are representative of the globally-averaged emergence timescales? Line 15-16: remove sentence that begins "Using". If not, then replace with… "allows for more equitable comparison of projections of CESM with those of lower sensitivity / higher variability models (Figs. 3 and 2)." [because we cannot define high-sensitivity as a 'bias' since we do not know the sensitivity of the Earth's climate yet]

Line 19-20: remove first 2 sentences, and begin with: "Broadly, temperature changes are…"

Line 27: "Model disagreement on the relative timing of TOE(o2) and TOE(T) is highest in the Atlantic…" A figure that bins the regions and computes regional SD across

the models for the quantity ToE(T)-ToE(O2) would be useful. Then you could make statements about the probabilities of certain regions following your hypothesis. For example, the Atlantic certainty does not, but others may. It would give some better spatial insight into where the distributions of Figure 8 are occurring.

Line 27-28: The discussion of the subtropics sounds like an extension of the discussion of the Atlantic. It should be made clear that you are now talking more broadly about the subtropics globally (I think) or still just discussing the Atlantic.

Page 11 Line 1: remove "it is striking that" and later correct to "typically show a decrease" Line 15: replace "noticeable" with "notable" Lines 1-15 are understandable but could benefit from some heavier rewording and condensing."

Line 17: "leading to relatively smaller changes in [O2]"

Line 21: rewrite to be specific: "anthropogenic change in temperature is detectable earlier than anthropogenic change in O2 most of the global ocean"

Page 12 Line 5: "trend" should actually be "change" Line 5-6: Rewrite "Both the magnitude of anthropogenic change and internal variability are model dependent, rendering the absolute year of ToE (strongly*) model-dependent*. Evaluating differences in absolute year of ToE, however, can obscure important model agreement upon the spatial patterns and progression of emergence within a multi-variable framework. We therefore introduce a new metric..."

*However, I note a recent NCC paper (Nijsse et. al., 2019) that argues that there is a correlation (and mechanistic relation) between climate sensitivity and decadal variability. Presumably if there is compensation between the two, then ToE could be relatively more robust across models than S or N.

Line 9: "within a model" is misleading – because this sounds as if the models are normalized by some external factor, say by taking "temperature of emergence" (the amount of global warming required for a signal in any chosen variable to emerge). I

think you could add a qualifier, and way "signal emerges relatively early or late relative to the signals global average in the given model."

Line 12: Replace short-coming with limitation or caveat

Line 14: Remove "Perhaps not surprising,"

Page 13 Line 16: Is it known why subtropical gyres in the Atlantic and the eastern equatorial Pacific have elevated noise? Please include. Line 23: Add word "spatial" before pattern. Lines 25-27: Rewrite: Even though the internal natural variability is low in the tropical regions, the O2 signal does not emerge from the noise, so is the signal."

Line 30: add "relatively" before "strong".

Line 35: change "highly" to "significantly"

Page 14 Line 1: remove "largely", if needed replace with "significantly" Line 2-3: Rewrite: "ToEs computed from CESM1.0 projections, for example, differ by many decades in absolute terms from other CMIP5 models, mostly due to a very weak internal natural variability." Line 3-4: Rewrite, something like "To extract valuable insights as to the relative spatial and temporal features of emergence across models and variables, we introduced..." However this notion is redundant with previous page so it could be excluded altogether or merged.

---

## Referee Comment (RC2) · Anonymous Referee #2 · 18 Oct 2019

This paper presents an analysis of the local time of emergence of an anthropogenic temperature and oxygen changes in the global oceans. In a recent study, the same authors (with the exception of Frolicher) used a single model to investigate the same topic. This paper went a step further by using an ensemble of Earth System models (ESMs) included in CMIP5.

The idea of using a single metric, the time of emergence (ToE) to determine the pint in time when the anthropogenic signal becomes larger than natural variability, is simple and appealing. The authors applied ToE to temperature and O2. Because ToE varies a lot among ESMs, they introduced the concept of relative ToErel by subtracting the global area-averaged ToE from ToE at each model grid point. Nevertheless, the results on ToE and ToErel would likely be sensitive to the threshold value (2) selected

in Equation (1) as well as the way how S (anthropogenic signal) and N (internal natural variations or background noise are calculated. Although similar calculations were reported in previous papers, the authors need to describe how S and N were calculated and examine the sensitivity or robustness of the model results. There are also questions why the same methodology can be used for different regions of the global oceans? Can you use the same methodology for the tropics and mid-latitudes?

The simple concept of ToE or ToErel also has its drawback, making it hard to interpret the model results. The authors provided little or no interpretations of the major models results (Figures 1-7). After reading the manuscript, I was left with an impression that it was a purely numerical exercise.

Some detailed comments:

(1) First paragraph in Section 3.1.1 on page 7. Why does ToErel (T) show early emergence in low latitudes and between 30o and 60o S and late emergence in the western tropical Pacific? Before jumping to ToErel, tell us the global mean ToE first. Why is there no emergence in the subtropical gyres of the Indian and the Pacific oceans?

(2) Second paragraph in Section 3.1.1 on page 7. Why is the spread among ESMs small in some regions but large in other regions?

(3) Third paragraph. Can you use individual model projections to obtain a quantitative estimate on the robust/uncertainty in estimating the mean ToE from the ESM ensemble?

(4) First paragraph in Section 3.1.2 on page 7. Why is ToErel (O2) relative homogeneous?

(5) Line 3 in the third paragraph in Section 3.3 on page 9. Large warming of more than ~4 Co is projected in the northern North Atlantic and round the subantarctic water. Can this projection be trusted? Many ESMs showed biases when simulating historical periods. Were these biases removed before the ToE analysis was applied?

[Figure]

(6) Section 3.4. It was a good idea to check changes in AOU in order to better distinguish the O2 and temperature signals. Can you check if ventilation of the thermocline indeed decreases in regions with decrease in [-AOU] rather than relying on cited references? The authors were on the right track here to get at the mechanisms but did not go far enough. Similar mechanistic analysis should be done to explain the other results.

(7) Second paragraph in Section 4 (page 12). Most ESMs do not have fine resolutions to simulate the oxygen minimum zones (OMZs) well. As the authors indicated, the ESMs diverge in their projections for the physical and biogeochemical changes in OMZs. Some models even showed an opposite trend to the observations in recent decades. This raised an important concern about the merit of even using such models to investigate ToE because they will lead to misleading results. Why didn't you remove those ESMs that did not capture the past changes?

Given these concerns, I cannot support the manuscript for publication in Biogeosciences in its present form, and recommend that the manuscript be returned for resubmission or major revisions.

---

## Referee Comment (RC3) · Anonymous Referee #3 · 23 Oct 2019

This paper focus on detecting the anthropogenic signals of both thermocline temperature and o2 (i.e. physical and biogeochemical properties) from a suite of CMIP5 models under the future projections. The study is (to some extent) based on the previous study of Hameau et al., (2019) extending to multi-model perspectives detecting the ToE of the thermocline temperature and o2 to assess the robustness of the results. The authors also introduce the relative ToE concept, results in reducing the inter-model spread compared to the traditional ToE and allows them to conduct more robust comparison.

In general I think it is important to aim on understanding changes in both physical and biogeochemical tracers together to better understand the resulting changes in marine ecosystems and combining multi-tracers could provide additional insights. I think the topic and contents of this study fits into the scope of the special issue in Biogeo-

sciences. However, I have comments on the current manuscript.

General Comments

1. Abstract: I suggest to include some discussion (possibly in the section 4) on following up the statement "... the detection of anthropogenic impacts become more likely when using multi-tracer observations" in the abstract. Combination of two tracers will definitely provide additional information on further implications from both physical and biogeochemical perspectives. Despite the fact that two of these properties emerge on different timescales, what would authors expect to see from (or should be aware of for monitoring) future multi-tracer observations?

2. I agree that the ToE comparison among the models are not straightforward and the advantage of relative ToE is to "reduce the inclusion model uncertainty in the metric" (as stated in section 3.2 in details). However, in section 3.4 (results on ToE comparison between the two variables), the author calculate the difference between the "absolute ToE" for each models. I thought this will still include more model bias (from global ToE, which is subtracted in relative ToE). Since the author introduced an improved ToE metric, it might be better to come up with a metric comparing "relative ToE" from the two variables. This might not be straightforward but can you think of further metrics based on comparing two relative ToEs? If authors think the this will not make a difference, please explain in more details.

3. Regarding to the terminology used in the manuscript, the "internal natural variability" and "natural variability" are mixed used in the manuscript. It is not always clear what exactly the terminology defines in this context. From what I understand, the internal natural variability meant here is the variability stemming from internal climate system (specific example will be ENSO, PDO etc.) and the natural variability includes the natural "external" forcing (such as volcanic eruptions) correct? It was mixture of terminology (particularly in the discussion, which I saw the two terms were inter-changeable in some sentences) and I suggest to clearly define the terminology in the beginning

and use the term in a consistent manner.

4. I suggest the authors to explain some of the statistics in more details in the method. The noise (N) is defined as the standard deviations from the pre-industrial control simulations from each models but did the author defined standard deviations based on the temporal standard deviation using full control simulation period? This may not be a huge difference but I assume periods differs among the model. Also, the CESM1 in this study uses the last millennium spinup but is this different from the preindustrial control simulations (I assumed yes)? I think introducing a schematic based on for example Figure A1 c) d) (or similar figure) will help explaining the N and S, and at which point you define the ToE used in this study in a more visualized way.

Specific Comments

- Page 2, L24-25: I suggest to cite one of the Oschlies review paper in addition to Cocco and Bopp's papers (underestimating the trend and variability of o2 in the model simulations).

- Page 4, Method, Earth system models section: I am guessing this will not affect much on the overall results but why did you use your own CESM1 (with different spin-up procedures) for multi-model comparison instead of using CMIP5 CESM? In addition, is the CESM1 used in this study the same as the one in the early Hameau et al., (2019)?

- Page 9, L1: What do you exactly mean by "combining climate sensitivity to anthropogenic forcing and natural variability in one metric"? I understand combining the anthropogenic forcing and natural variability part but I was not fully sure about the climate sensitivity statement.

- Page 12, L18: "an increasing ventilation" (following Gnanadesikan et al., 2007): Strictly speaking I would not state "an increasing ventilation" but it is more of a consequence of reduced upwelling as discussed in Gnanadesikan et al., 2007.

- Figure 1. I understand from the Figure 1 that the SD reduces for the relative ToE

[Figure]

but I also have some impression that two metrics could still give similar information. It might help to show additional map of ToE SD difference between Figure 1 (b) and (c) for example to show the bias (spread) reduction using this metric.

- Figure 6. For consistency, I suggest to used the same hatching as the previous figures to show the regions that one of the variables has not emerged by 2099 rather than saturated colors.

- Figure 8. I like this summary figure aiming on incorporating emergent signals of both thermocline temperature and o2, along with AOU information (mainly indicating the water mass age information). Minor things on this, I think the x-axis is supposed to be -AOU (it puzzled me for a moment) and x-axis label should be corrected.

References:

[1] Hameau, A., Mignot, J., and Joos, F. (2019), Assessment of time of emergence of anthropogenic deoxygenation and warming: insights from a CESM simulation from 850 to 2100 CE, Biogeosciences, 16, 1755–1780, https://doi.org/https://doi.org/10.5194/bg-16-1755-2019.

[2] Oschlies et al., (2017), Patterns of deoxygenation: sensitivity to natural and anthropogenic drivers. Philosophical Transactions of the Royal Society A: Mathematical, Physical and Engineering Sciences, 375 (2102). p. 20160325. DOI 10.1098/rsta.2016.0325.

[3] Gnanadesikan, A., Russell, J. L., and Zeng, F.: How does ocean ventilation change under global warming?, Ocean Science, 3, 43–53.

---

## Author Comment (AC1) · 19 Nov 2019

We thank the reviewers for assessing this manuscript and for their time and effort. The useful comments are much appreciated and helped to improve the presentation of our results. Please find attached to this reply a revised manuscript where text changes are highlighted.

Please also note the supplement to this comment: https://www.biogeosciences-discuss.net/bg-2019-339/bg-2019-339-AC1-supplement.pdf

---

## Author Comment (AC3) · 19 Nov 2019

**Reply to review comments**

We thank the reviewers for assessing this manuscript and for their time and effort. The constructive comments are much appreciated and have improved the manuscript. The original review comments are given below in black, our reply in blue, and quotes from the revised manuscript in gray.

Please find attached to this reply a revised manuscript where text changes are highlighted.

**1 Sarah Schlunegger Referee #1**

**1.1 General**

Hameau et. al. provides a multi-model (CMIP5) assessment of the relative timing of emergence of anthropogenic change in thermocline oxygen and thermocline temperature. To facilitate multi-model assessment of time of emergence (ToE), they provide a new metric, relative ToE. They find that for most of the global ocean changes in temperature emerge prior to changes in oxygen, however for some regions, changes in oxygen emerge prior to changes in temperature. Anthropogenic changes in oxygen emerge prior to temperature in regions where reduced solubility and ventilation work in tandem to reduce oxygen concentrations. In these locations, reduced ventilation also slows the propagation of anthropogenic warming signals from the upper ocean into the ocean interior, further contributing to the delay between emergence of thermocline oxygen and temperature. This paper represents an important contribution to the growing body of mechanistic interpretations of emergence timescales in the global ocean. Additionally, it confronts the known challenge of model-intercomparison with provision of a simple, yet powerful new metric, relative ToE. I recommend its publication in Biogeosciences, but only after minor-to-moderate revision of the text and inclusion of an additional figure to better visualize the regions for which thermocline oxygen robustly emerges prior to temperature.

Sincerely, Sarah Schlunegger, PhD Princeton University Program in Atmospheric and Oceanic Science
Thank you for your general support and your constructive comments.

**1.2 Specific:**

1. Improve Abstract In the abstract, I would suggest the following rewording to replace some of lines 7-10: Changes in thermocline oxygen emerge prior to changes in temperature because oxygen declines occur due to the confluence (or additivity) of both reduced solubility and ventilation. Otherwise, the abstract could be further streamlined and focused. Feel free to borrow from my summary above.
Thank you very much for the suggestion. The abstract has been revised and streamlined accordingly. Please see the revised manuscript draft at the end of this document.

2. Improve motivation/framing of the introduction Page 3 frames the question of "Which emerges first, physical or biogeochemical variables?" However, this is an ill-formed question, as the chronology of emergence has already been documented to not follow strict temporal separation of physical and biogeochemical variables. For example, Rodgers et al., 2015 evaluates ESM projection of 4 variables and find the following general emergence sequence: pH, SST, O2, NPP – biogeochemical, physical, physical, biogeochemical. Schlunegger et al., 2019, evaluates ESM projection 20 physical/biogeochemical variables, finding their emergence timescales are separated by their association to either chemical (gas-exchange) or physical (warming)

impacts of climate change on the ocean, and NOT to whether the variable itself is physical or biogeochemical. Variables that are first-order impacted by increasing atmospheric pCO2 and increased gas-exchange, like DIC, pH etc., will emerge the most rapidly (carbonate-chemistry related biogeochemical). Later, as atmospheric warming propagates into the ocean (physical) warming signals emerge, and quickly after solubility induces changes (like O2sat). Even later, circulation eventually adjust (physical), thereby altering nutrient supply (biogeochemical), and subsequently primary production (biogeochemical), export production (biogeochemical), etc. The broad order of emergence could then be described as 1. Carbonate-chemistry related biogeochemical and biology (hard tissue pump), 2. Surface temperatures and solubility-related variables (like O2), 3. Ocean dynamics, 4. Biogeochemical impacted by ocean dynamics (nutrients, production).

We have re-written the question. We now emphasize that we are concerned with the detection of the warming and oxygen signals in the ocean interior, as opposed to the surface, and in variables that are routinely observed versus less frequently measured. The text now reads:

Beyond the combined impact of physical and biogeochemical changes, an interesting question is whether anthropogenic changes in the ocean interior are first detectable in variables that are routinely and frequently measured such as temperature (T) or in variables with a relatively low observational coverage but potentially high impact for ecosystems such as $O_2$ (Joos et al., 2003). The answer may have implications for measurement strategies to detect anthropogenic changes in subsurface waters as well as for the impacts of physical and biogeochemical change on marine life. For the surface ocean, earlier studies (Keller et al., 2014; Rodgers et al., 2015; Frölicher et al., 2016; Schlunegger et al., 2019) showed that the anthropogenic signals of pH and pCO2 emerge earlier than sea surface temperature and $O_2$ change and earlier than productivity changes. Changes in surface $O_2$ are tightly coupled to temperature-driven solubility changes and $O_2$ varies hand in hand with sea surface temperature and the two signals emerge typically concomitantly. Regarding the ocean interior, the sequence of emergence for $O_2$ and T is less clear. Global warming increases surface ocean temperature, which tends to reduces $O_2$. On the other hand, $O_2$ is also influenced by non-thermal processes, such as respiration and the redistribution by ocean circulation and mixing. Respiration of organic matter in the ocean interior may have a larger influence on $O_2$ change than temperature-driven solubility change in a more stratified and less ventilated ocean. One could therefore expect that, under global warming, the combined effect of increased $O_2$ consumption and decreased $O_2$ solubility will accelerate the $O_2$ depletion in subsurface waters and that $O_2$ may be detectable before the warming reaches that layer.

3. Additional methodological explanation required Explanation of how S is computed, and its statistical properties relative to N, should be addressed more clearly. For instance, there are broadly two distinct usages of the term "time of emergence" – the first is to define the point in time at which the ocean state/variable is distinctly out of the range of the pre-industrial state, the second, is to define the point in time at which a forced trend is outside the range of how large natural trends are likely to be. I infer that the first meaning is used here, but that should be made clear.

We thank the reviewer for this pointer. Reviewer 2 raised a similar question. Therefore we have extended the paragraph related to the ToE methodology. In addition, a figure is now provided in the appendix to increase clarity (Fig. A1). The following sentence was also added:

Here, ToE represents the moment in time at which the ocean state becomes distinct from the preindustrial state.

4. Potentially reordering the figures to improve narrative flow. Another immediately odd thing that I see is that Figure 1a and 1b at first glance oppose the main hypothesis of the manuscript. This is just visually the case because O2 has higher model disagreement and T and therefore more area where at least 2 of the models think emergence does not occur this century. However, I know many (mostly senior) scientists who just read the abstract, skim the method and the look at the figures. From this alone, the figures do not tell your story until Figure 6. In fact, Figure 1 is somewhat irrelevant to the stated hypothesis in the introduction. The order of figures could therefore be revised to Figure 2, 3, 6, 7, 8, 1,4,5. With the text also reordered.

We thank the reviewer for this suggestion. We have We have now added a new figure in the discussion that is summarizing our main findings regarding early emergence of T versus $O_2$. We also formulated the abstract more concisely. We feel that these changes will convey our main message to all readers.

5. Additional figure to better visualize regions of ToE(O2) < ToE(T) These regions should be better visualized in the paper, potentially in a synthesis map showing locations of ToE(O2)<ToE(T) robust across models (could use the 7/9 threshold for example) and/or a figure/bargraph for which regions are aggregated and the average relTOE(O2) vs relTOE(T) is ploted for each model. So the figure would have regions on the X axis and ToE(O2)-ToE(T) on the Y axis. From there the reader could see the spread in the models as well as pick out regions where it is most likely for ToE(O2) to be significantly shorter than ToE(T).

Thank you for the suggestion. We have added a "binary" type map (Fig. 9) that shows the regions in blue/brown where seven out of nine models simulate emergence of oxygen/temperature before temperature/oxygen. We have added following text to the Discussion section:

On average across the nine models, an area covering $35\pm11$ % (Fig. 6) of the global thermocline shows emergence in $O_2$ change before temperature change. Yet, the exact locations of relatively early emergence of $O_2$ differ across models. Hence, the regions where at least seven out of the nine models show consistently an earlier emergence of $O_2$ than T is smaller and amounts to 17 % of the global thermocline area. As shown in Figure 9 (blue areas) the $O_2$ signal emerges consistently in at least seven models before the T signal in parts of the Pacific subtropical gyres, the Southern Ocean and the southeast Indian Ocean.

**1.3 Additional Technical/Editorial Corrections**

**Page 1**

Line 3: remove word "as" .
Done and acknowledged.

Line 12-14: Simplify sentence with: To normalize across disparate trends and variability of the CMIP5 ensemble, we compute the local ToE relative to the global mean ToE within each model.
Modified as suggested

**Page 2**

Line 4-5: correct to remove "and". . . adversely affect marine organisms, ecosystems, and the services they provide.
Modified as suggested

Line 6: remove "a": . . .experienced significant warming
Modified as suggested

Line 9: Make "scale" plural. . .on regional to local scales
Modified as suggested

Line 13: Slightly awkward jump to discussion of ESMs.
The sentence has been moved. See comment on page 4, line 25)

Line 17: Remove first phrase, "Concomitant with ocean warming" and just start with "Observation-based studies. . ." The mechanisms of observed oxygen decline is subsequently explained, so it needn't be partially explained initially.
Modified as suggested

Lines 19-21: Break into 2 sentences and add context. "In subsurface waters, oxygen concentration is also affected by ventilation, remineralization of organic matter and air-sea disequilibrium. In the contemporary ocean, oxygen decreases are mostly dominated by a reduction in ventilation and increased consumption of oxygen during remineralization (references)."
We split the sentence as suggested. However, we find it confusing for a general reader to say that changes in air-sea disequilibrium affect $O_2$ in subsurface waters. Further, most model studies also suggest that changes in ventilation are, at least on larger scales as considered in this study, more important than changes in

export production of organic material. The text reads now:

Increased surface temperature reduces oxygen solubility, limiting atmospheric oxygen dissolution into the upper ocean. In subsurface waters, oxygen concentration is also affected by changes in ventilation and the remineralisation of organic matter. In the contemporary ocean, oxygen decreases in the interior are mostly dominated by a reduction in ventilation with a smaller role for changes related to the production of organic matter, $O_2$ solubility, and air-sea equilibration of $O_2$ in surface waters (Bopp et al., 2002; Plattner et al., 2002; Bopp et al., 2017; Tjiputra et al., 2018; Hameau et al., 2019).

Line 21: change to. . . "The largest oxygen declines are located"

Modified as suggested

Line 23: define "late" industrial period.

The sentence reads now:

The largest oxygen declines are located in the Pacific Ocean (equator and northern hemisphere) and the Southern Ocean. However, observations are relatively sparse and only start in the second half of the 20th century.

Line 23-24: change to. . . "Therefore, it is challenging to distinguish human-caused trends from natural variations in the observational record of ocean O2."

Modified as suggested

Line 24: remove "also". . . Modelling studies agree on the sign of . . .

Modified as suggested

Line 24-26 could be combined instead with 13-16 and form their own paragraph discussing the current state of modeling O2 trends.

The description of models studies for temperature and oxygen are now combined in one paragraph.

Global climate models, such as the Earth system models that participated in phase 5 of the Coupled Model Intercomparison Project (CMIP5) reproduce the long-term trend in global ocean heat content over the last 50 years when uncertainties of observation-based estimate and internally generated natural variability are taken into account (Frölicher and Paynter, 2015; Cheng et al., 2019). Modelling studies agree on the sign of oceanic $O_2$ changes, but likely underestimate the magnitude of loss (Cocco et al., 2013; Bopp et al., 2013; Oschlies et al., 2017). In particular in the tropical regions, models are not able to reproduce observed $O_2$ decrease in equatorial low-oxygen zones (Stramma et al., 2008; Cocco et al., 2013; Cabré et al., 2015).

Line 32: include reference, " (e.g. Deutch et. al., 2015)"

Reference included as suggested.

**Page 3**

Line 11-12: rewrite. . ."is critical to understanding contemporary O2 and temperature changes."

Modified as suggested

Line 17-29: rewrite "One study, (Hameau et al., 2019), uses a single model [ which model?], to investigate ToE of temperature in the thermocline, finding that anthropogenic ocean warming emerges much earlier than the O2 signal in low and midlatitude regions. Delayed emergence of changes in O2 is due to the opposing effects of decreases in O2 solubility and O2 consumption. In the high latitudes and the Pacific subtropical gyres, deoxygenation emerges before ocean warming in [ name model]. This occurs because decrease in oxygen solubility are reinforced by increased O2 consumption, leading to strong O2 depletion However, it is unknown if this single-model result is robust across a suite of different Earth system model simulations. Here, we conduct a multi-model study to more broadly test the hypothesis that anthropogenic deoxygenation in the thermocline emerges prior to anthropogenic warming. Since the primary objective is

to test the consistency across models of the order of emergence (deoxygenation prior to warming) within a single model, we introduce and use a relative ToE to conduct the intercomparison, rather than the absolute year of ToE. We define relative ToE as. . . ”

Modified as suggested

**Page 5**

Lines 1-10: Over what time frame is S estimated? The texts read "Enting, 1987"... Ending in 1987? At first I thought it was a typo but now I see it is a reference. To clarify, maybe state that S is a time series that extends from pre-industrial to 2100 (If I am understanding correctly). Secondly, with what "fitting?" –linear fit, polynomial, etc.? Perhaps the word "fit" should be excluded, if it is in fact meant to just say that the time-series is low-pass filtered.

According to your comment page 2, line 31, the corresponding paragraph has been improved and clarified. Moreover, it is now stated that the period of the signal $S$ extends the entire period of the simulation $(1860 - 2099)$.

Line 20/21: change request to require and is to are: "We therefore require that ToE values are defined for at least seven out of nine models to compute the multi-model statistics (median and spread)."

Modified as suggested

Line 24: New section, entitled "Separating mechanisms of oxygen change" or something like this.

Modified as suggested

Line 24-25: Potentially rewrite: To diagnose processes driving the simulated changes in ocean O2, the direct thermal/solubility component of change (O2sat) can be isolated from the total O2 change. The residual, Apparent Oxygen Utilization (AOU), represents the summation of all non-thermal changes, including those resulting from changes in ventilation and remineralization.

Modified as suggested

**Page 9**

The early emergence of T in the Eastern Equatorial pacific should be discussed, and if possible, mechanistically explained. At first glance, it is surprising that an area of such interannual variability at the surface can have relatively little variability at intermediate depths.

The strong variability in the Eastern Pacific upwelling system does not reach the $200 - 600$m layer. As shown in Fig. 1, internal variability is confined between the surface and 200 m. Therefore, the temperature increase at $200 - 600$ m depth range are detectable relatively rapidly.

The region in the Eastern Tropical Pacific is now explicitly discussed in the main text.

The combination of a strong signal and small variability results in early detection of the changes. This is the case in the Southern Ocean at 45° S (in the Atlantic and Indian regions; Fig. 1a), where the anthropogenic warming is strong (up to 4 ° C; Fig. 4b) but the internal variability is relatively small (0.1 ° C to 0.3 ° C; Fig. 4a). However, early emergence of anthropogenic changes can also occur when the signal is relatively small, if the variability is even smaller. This is the case in the tropical oceans such as in the Arabian Sea and the equatorial Atlantic, where water masses warm modestly (up to 1.5 ° C), but vary naturally between 0.1 ° C and 0.2 ° C only. It is also the case in the eastern equatorial Pacific, where the early emergence arise from the very weak internal variability in the thermocline, although, the temperature increase ($\sim$0.80 ° C) is also relatively weak. In this region, the substantial variability in $O_2$ and T is largely confined to the top 200 m. No emergence by the end of the 21st century, such as simulated in the subtropical gyres of the Indian and Pacific oceans, results from a relatively weak signal combined with a relatively strong variability in these regions.

[Figure]

Figure 1: Internal temperature variability simulated by the CESM model under preindustrial conditions (top left) at the surface, (bottom left) for the averaged layer 200 – 600m, (top right) along the meridional 110 ° W (illustrated by the blue dashed line in the left panels). The bottom right panel shows the temperature increase by the end of the 21st century along the same longitude.

Line 20: remove word 'typically'
Modified as suggested

**Page 10**

₅ Line 11: remove word 'surprisingly'
Modified as suggested

Line 12: what is the confidence interval referenced? Is this references figure 1e, the multi-model spread? If the medium relTOE is 10 and the spread is also 10, then what does "confidence interval" mean in this
₁₀ context? And in that case, a confidence interval value of 1 is not "high". Usually confidence interval terminology is used to describe the probability of rejecting of a null hypothesis. The implicit hypothesis of the statement would be that in the CMIP5 ensemble, this region's relTOE($O_2$) does not significantly differ from the global average TOE($O_2$). Potentially a better way to convey this is to state that the multi-model ensemble agrees that the North Pacific represents a region for which emergence timescales are representative
₁₅ of the globally-averaged emergence timescales?
We agree with your comment on the use of "confidence interval". The sentence reads now:
However, ToE $_{rel}$($O_2$) in this region is within ∼10 years (Fig. 1d), with a relatively low spread (±10 years) compared to ToE $_{abs}$($O_2$) (±30 years).

₂₀ Line 15-16: remove sentence that begins "Using". If not, then replace with ... "allows for more equitable comparison of projections of CESM with those of lower sensitivity / higher variability models (Figs.3 and 2)." [because we cannot define high-sensitivity as a 'bias' since we do not know the sensitivity of the Earth's climate yet]
Modified as suggested and word biases replaced with model-model differences
₂₅ The ToE$_{rel}$ allows the comparison of ToE resulting from CESM output with the results from the 8 models in spite of these model-model differences (Figs. 2 and 3).

Line 19-20: remove first 2 sentences, and begin with: "Broadly, temperature changes are. . ."
The two first sentences have been removed. The beginning of the paragraph reads now:
₃₀ In general, temperature changes are detectable before $O_2$ changes in around 64±11 % of the thermocline

Line 27: "Model disagreement on the relative timing of TOE(o2) and TOE(T) is highest in the Atlantic. . ."
A figure that bins the regions and computes regional SD across the models for the quantity ToE(T)-ToE(O2) would be useful. Then you could make statements about the probabilities of certain regions following your
₃₅ hypothesis. For example, the Atlantic certainty does not, but others may. It would give some better spatial insight into where the distributions of Figure 8 are occurring.
As suggested, we have added a "binary" type map (Fig. 9) that shows the regions where at least 7 models project earlier emergence in oxygen (blue) or temperature (brown)

₄₀ Line 27-28: The discussion of the subtropics sounds like an extension of the discussion of the Atlantic. It should be made clear that you are now talking more broadly about the subtropics globally (I think) or still just discussing the Atlantic.
The sentence has been updated and reads now:
Model results for the Atlantic subtropical gyres are mixed. Some models suggest $O_2$ changes to be detectable
₄₅ earlier than T changes (HadGEM2 and the IPSL family), whereas in other models the $O_2$ signal does not even emerge.

**Page 11**

Line 1: remove "it is striking that" and later correct to "typically show a decrease"
₅₀ Modified as suggested

Line 15: replace "noticeable" with "notable"
Modified as suggested

5    Lines 1-15 are understandable but could benefit from some heavier rewording and condensing."
The paragraph has been rewritten and reads now:
Regions with early emergence of anthropogenic $O_2$ compared to T show typically a decrease in [-AOU]
(Fig. 6 versus Fig. 7), whereas regions with early emergence of T compared to $O_2$ show typically an increase
in [-AOU]. For example, [-AOU] is decreasing in 77±8 % of the areas with early emergence of $O_2$, while only
10   22±8 % of these regions show an increase in [-AOU] (Fig. 8; blue). In most regions where T is emerging
before $O_2$ (Fig. 8; green), [-AOU] is increasing (62±12 %). A decreasing trend in [-AOU] is indicative of
a reduced ventilation induced by upper ocean warming and increased stratification (e.g. Capotondi et al.,
2012). A more sluggish ventilation slows the supply of $O_2$ from the surface to the ocean interior. Conse-
quently, thermocline $[O_2]$ and [-AOU] are both decreasing. This leads to a strong and thus early detectable
15   anthropogenic deoxygenation. In addition, a more sluggish ventilation slows the penetration of the an-
thropogenic warming signal from the surface to the interior, and similarly the penetration of the thermally
driven $O_2$ signal ($[O_{2,sol}]$). The detection of the temperature changes is thus delayed compared to AOU and
$O_2$. There are some exceptions to this relationship between [-AOU] and the earlier emergence of $O_2$ than T.
For example, $O_2$ change emerges before warming in the GFDL model around 30° S and 120° W, although
20   [-AOU] is increasing in this region. However, warming is emerging very late as the GFDL models simulate
weak warming and even some cooling (Fig. S4) in this part of the thermocline. Thus, in this special case,
the early emergence of $O_2$ relative to T is due to the absence of large warming in a region with notable
temperature internal variability.

25   Line 17: "leading to relatively smaller changes in [O2]"
Modified as suggested

Line 21: rewrite to be specific: "anthropogenic change in temperature is detectable earlier than anthro-
pogenic change in O2 most of the global ocean"
30   Modified as suggested

**Page 12**

Line 5: "trend" should actually be "change"
The sentence reads now:
35   Using ToE as a metric allows for the assessment of anthropogenic changes by comparing the magnitude of
the human-induced changes with the magnitude of internal variability.

Line 5-6: Rewrite "Both the magnitude of anthropogenic change and internal variability are model depen-
dent, rendering the absolute year of ToE (strongly*) model-dependent*. Evaluating differences in absolute
40   year of ToE, however, can obscure important model agreement upon the spatial patterns and progression
of emergence within a multi-variable framework. We therefore introduce a new metric. . ."
Modified as suggested

*However, I note a recent NCC paper (Nijsse et. al., 2019) that argues that there is a correlation (and
45   mechanistic relation) between climate sensitivity and decadal variability. Presumably if there is compensa-
tion between the two, then ToE could be relatively more robust across models than S or N.
Thank you for pointing to this publication. We have added to following sentence in the discussion.
Nijsse et al. (2019) suggest that the magnitude of simulated decadal variability and climate sensitivity might
be correlated. They suggest that models with a high climate sensitivity tend to simulate a high decadal
50   variability. This may imply a compensation between the simulated signal and noise on the decadal scale.

Line 9: "within a model" is misleading – because this sounds as if the models are normalized by some external factor, say by taking "temperature of emergence" (the amount of global warming required for a signal in any chosen variable to emerge). I think you could add a qualifier, and way "signal emerges relatively early or late relative to the signals global average in the given model."

We changed "within a model" by "for a given model". The proposed sentence tends to repeat the definition of relative ToE. But here, we want to highlight the general benefit of using it. The sentence reads now:
 Absolute years of emergence are thus not considered by this metric and it only illustrates whether a signal emerges relatively early or late for a given model.

Line 12: Replace short-coming with limitation or caveat
Modified as suggested

Line 14: Remove "Perhaps not surprising,"
Modified as suggested

**Page 13**

Line 16: Is it known why subtropical gyres in the Atlantic and the eastern equatorial Pacific have elevated noise? Please include.
The Eastern Equatorial Pacific shows a low variability in the thermocline. We corrected this wrong statement. We did not investigate in detail why there is high variability in the subtropical gyres of the Atlantic. We suspect that this is related to a large variability in the winds which force these gyres. The text now reads:
Exceptions are the subtropical gyres in the Atlantic, where it takes approximately two additional decades to detect the temperature changes, mainly because of the relatively large internal variability there.

Line 23: Add word "spatial" before pattern.
Modified as suggested

Lines 25-27: Rewrite: Even though the internal natural variability is low in the tropical regions, the O2 signal does not emerge from the noise, so is the signal."
The proposed modification seems confusing to us. However, we revised the sentence for clarity:
Although internal natural variability is low in the tropical regions, the $O_2$ signal does not emerge by 2100. This is because the projected changes are also small.

Line 30: add "relatively" before "strong".
Modified as suggested

Line 35: change "highly" to "significantly"
Below the updated sentence:
For example, the transient climate response of the individual models and therefore the ocean heat uptake, thermocline warming and deoxygenation can be substantially different (Bopp et al., 2013).

**Page 14**

Line 1: remove "largely", if needed replace with "significantly"
Below the updated sentence:
In addition, the simulated internal variability considerably differs across models (e.g. Resplandy et al., 2015; Frölicher et al., 2016).

Line 2-3: Rewrite: "ToEs computed from CESM1.0 projections, for example, differ by many decades in absolute terms from other CMIP5 models, mostly due to a very weak internal natural variability."
The sentence reads now: ToEs computed from CESM1.0 projections, for example, differ by many decades

in absolute values from other CMIP5 models, mostly due to a very weak internal variability.

Line 3-4: Rewrite, something like "To extract valuable insights as to the relative spatial and temporal features of emergence across models and variables, we introduced. . ." However this notion is redundant with previous page so it could be excluded altogether or merged.
The sentence has been modified as suggested

**2 Anonymous Referee #2**

**2.1 General**

This paper presents an analysis of the local time of emergence of an anthropogenic temperature and oxygen changes in the global oceans. In a recent study, the same authors (with the exception of Frolicher) used a single model to investigate the same topic. This paper went a step further by using an ensemble of Earth System models (ESMs) included in CMIP5. The idea of using a single metric, the time of emergence (ToE) to determine the pint in time when the anthropogenic signal becomes larger than natural variability, is simple and appealing. The authors applied ToE to temperature and O2. Because ToE varies a lot among ESMs, they introduced the concept of relative ToErel by subtracting the global area-averaged ToE from ToE at each model grid point.

Nevertheless, the results on ToE and ToErel would likely be sensitive to the threshold value (2) selected in Equation (1) as well as the way how S (anthropogenic signal) and N (internal natural variations or background noise are calculated).

We agree with the reviewer that methodological choices affect results for ToE in detail as shown earlier (Rodgers et al., 2015; Frölicher et al., 2016; Hameau et al., 2019). Our main conclusion - the signal of anthropogenic $O_2$ is emerging earlier than the anthropogenic warming signal in many ocean regions - was found to be robust regarding specific methodological choices applied to results from a single model (CESM) (Hameau et al. (2019)). Here, we find that our main conclusion is also robust across the different ESMs and thus for a range of different noise levels. The following text has been added in the introduction to clarify this:

The following paragraph has been added to the discussion:

Although the generic definition of ToE is under consensus, the methodologies applied to estimate ToE differ in the published literature (IPCC (2019)) as mentioned in the introduction. Depending on the spatial and temporal scale of a given variable, the threshold for which emergence is defined and the reference period applied, the absolute value of ToE can differ. In addition, the ToE also depends on the definition of the background variability, here acting as noise (Hameau et al., 2019). Estimating the background noise as the standard deviation (SD) of the internal chaotic variability from the control simulation (Frölicher et al., 2016), or as the SD of the variability from the industrial period (after removing anthropogenic trends; Keller et al., 2015; Henson et al., 2016) result in earlier ToE for both $O_2$ and T as when estimating the noise from the total (internal and externally-forced) natural variability over the last millennium. Yet, the finding that anthropogenic $O_2$ change emerges before anthropogenic warming in large ocean regions is robust across investigated choices. The anthropogenic signal is frequently computed as a linear trend over a few decades (Rodgers et al., 2015; Henson et al., 2017; Tjiputra et al., 2018). However, the resulting slope depends on the time window used to calculate the linear trend. Therefore, Hameau et al. (2019) use a low-pass filtered output to estimate the signal.

Although similar calculations were reported in previous papers, the authors need to describe how S and N were calculated and examine the sensitivity or robustness of the model results. There are also questions why the same methodology can be used for different regions of the global oceans? Can you use the same methodology for the tropics and mid-latitudes?

Thank you for pointing out that we do not provide sufficient explanations for those readers not familiar with the ToE concept. The method section has been extended and an additional figure has been added to the appendix (Fig. A1). The sensitivity of results to methodological choices is explored and quantified in our previous publication (Hameau et al., 2019) as well as in IPCC (2019) - Chap5 - Box5.1 entitled: "Time for Emergence and Exposure to Climate Hazards"; please see our answer to the previous comment (page 11, line 14). In this manuscript, the robustness of relative and absolute ToE estimations across the model range is discussed in Sect. 3.1 and illustrated by Fig. 1 (middle and lower panels).

Yes, the same ToE method can be applied to any time series and to different ocean regions as is it computed in each grid point. However, the definition of ToE could differ among regions because of specific behaviour of the considered variability with depth and time for example. Nevertheless, our objective here is to investigate a specific methodology across models and thus we leave this latter point for future studies.

The following technical explanation is now given in the method section:

We define the absolute ToE as the first year when the anthropogenic signal $S$ becomes equal or larger than twice the noise of internal variability $N$ (Eq. 1; following Hameau et al., 2019; Fig. A1). The threshold is set to two in order to distinguish the signal from the noise at 95 % confidence level. Annual $O_2$ and T data are first averaged over the thermocline (200 – 600 m) at each grid point of the horizontal grid and local $S$ and $N$ are computed from these depth-averaged values for each model, variable and (horizontal) grid-point. Annual anomalies are calculated relative to the preindustrial period (1860 – 1959)....

The background noise, $N$, is computed as one standard deviation (SD) of $O_2$ and of T from the annual preindustrial control output. Although, the length of the control simulations differ between models, the entire duration of the control simulation is considered for each model to estimate the background noise....

The annual output of the forced, transient simulation (1860 – 2099) is smoothed by a low pass spline filter (Enting, 1987) to estimate $S$ for each (horizontal) grid point in the thermocline. The cut-off period of the spline is set to 80 years to remove decadal to multi-decadal variations (e.g. associated with internal variability). The signal $S$ is then the value of the spline at each point in time.

The simple concept of ToE or ToErel also has its drawback, making it hard to interpret the model results. The authors provided little or no interpretations of the major models results (Figures 1-7). After reading the manuscript, I was left with an impression that it was a purely numerical exercise.

We disagree with the reviewer regarding the importance of the ToE concept and of this study, and refer to the general comments of reviewer #1 (see page 1, line 11) and #3 (see page 17, line 7). We agree that it is difficult to interpret ToE and $ToE_{rel}$ in a mechanistic way even within a single model. ToE reflects the ratio between the magnitude of change and the magnitude of variability - two quantities that are, at least partly, influenced by different processes. Providing mechanistic explanations in this multi-model study is beyond the scope of this work. We note that we quantify both magnitude of anthropogenic change and variability individually and that we distinguish between solubility-driven and remineralization-driven $O_2$ changes. We show that anthropogenic deoxygenation emerges before anthropogenic warming in about a third of the global thermocline. This, together with the finding that human-caused $O_2$ changes leave the bounds of natural variability in many regions is likely relevant to assess the risks associated with anthropogenic greenhouse gas emissions. We conclude that monitoring biogeochemical variables such as $O_2$ will help to better identify environmental risks.

**2.2 Some detailed comments:**

**(1)**

First paragraph in Section 3.1.1 on page 7. Why does ToErel (T) show early emergence in low latitudes and between 30° and 60° S and late emergence in the western tropical Pacific?

The driver of early/late emergence are discussed in details in Sect 3.3. The corresponding sentences have been updated to clarify this:

The combination of a strong signal and small variability results in early detection of the changes. This is the case in the Southern Ocean at 45° S (in the Atlantic and Indian regions; Fig. 1a), where the anthropogenic warming is strong (up to 4 ° C; Fig. 4b) but the internal variability is relatively small (0.1 ° C to 0.3 ° C; Fig. 4a)....

...However, early emergence of anthropogenic changes can also occur when the signal is relatively small, if the variability is even smaller. This is the case in the tropical oceans such as in the Arabian Sea, the equatorial Atlantic and the western equatorial Pacific, where water masses warm modestly (up to 1.5 ° C), but vary naturally between 0.1 ° C and 0.2 ° C only....

In order to guide the reader, the beginning of the result section reads now:

We start by discussing the multi-model median and spread of relative ToE estimates for potential temperature (Fig. 1a, b) and dissolved oxygen (Fig. 1d, e) changes in the thermocline (200 – 600m). An analysis of the roles of internal variability and anthropogenic change ToE and why anthropogenic change is detectable early or late is presented in Sect 3.3.

Before jumping to ToErel, tell us the global mean ToE first.

The following text is added in section 3.1.1. and 3.1.2:

ToE values for individual horizontal grid cells are globally averaged to obtain an area-weighted global mean ToE for the thermocline and each model. These global mean values range between year 1963 and year 2033 for the nine models (see subtitles in Fig. 2). These global differences between models are removed by definition in ToE $_{rel}$. The multi-model median of ToE $_{rel}$(T) shows early emergence in low latitudes and between $30^o$ S and $60^o$ S, ...

As for temperature, a large range in absolute ToE is found with globally-averaged ToE($O_2$) ranging between the year 1991 and 2046 for the nine models (see subtitles in Fig. 3).

Why is there no emergence in the subtropical gyres of the Indian and the Pacific oceans?

No emergence of warming until the end of the 21st century result from the combination of relatively low anthropogenic warming and relatively strong internal temperature variability. As discussed in Sect 3.3, high temperature variability is simulated in the subtropical thermocline. Subtropical gyres are regions of vertical subduction and thus active wind ventilation (Pedlosky, 1996). This induces relatively large variability in these regions and may explain late emergence. Differences in vertical profiles of temperature and salinity and in effective thermocline depths among the different basins may explain the specific behavior of the Pacific and Indian oceans as compared to the Atlantic in our diagnostics. We added the following sentence:

No emergence by the end of the 21st century, such as simulated in the subtropical gyres of the Indian and Pacific oceans, results from a relatively weak signal combined with a relatively strong variability in these regions.

[Figure]

Figure 2: Scatter plots between the multi-model spread in relative (top panel) ToE(O$_2$) and (lower panel) ToE(T) and the multi-model median of (a) relative ToE, (b) anthropogenic change (1860-1959 – 2070-2099) and (c) internal chaotic variability. Each point represents a grid cell in the thermocline.

**(2)**

Second paragraph in Section 3.1.1 on page 7. Why is the spread among ESMs small in some regions but large in other regions?

This is an interesting question. We are not able to answer this question. We do not find a clear relationship between the multi-model spread in ToE and the multi-model median in ToE, or the multi-model median of the anthropogenic signal, or the multi-model median of the variability for both T and O$_2$ as evidenced by the scatter plots in Fig. 2 of this reply.

**(3)**

Third paragraph. Can you use individual model projections to obtain a quantitative estimate on the robust/uncertainty in estimating the mean ToE from the ESM ensemble?

Yes, please see Fig. 1 and related discussion in the manuscript. As usual in multi-model analyses, individual variables, here ToE and $ToE_{rel}$, are computed first for each individual model. Then, additional metrics are computed including the multi-model median and the multi-model spread. The interquartile ranges for ToE and $ToE_{rel}$ for T and $O_2$ are shown in Fig. 1b and e. The interquartile range represent a measure of model uncertainty. The following text is added to the method section to clarify this:

$S$, $N$, ToE and $ToE_{rel}$ are first computed from the annual output for each model and at each (horizontal) grid cell. Then, multi-model median and spread (interquartile range) of the multi-model estimations are computed from the model ensemble. The median represents a "best" estimate and the interquartile range a measure of model uncertainty. Uniform weights are applied to each model configuration to compute these statistics. ..

**(4)**

First paragraph in Section 3.1.2 on page 7. Why is ToErel (O2) relative homogeneous?

We thank the reviewer for pointing out this misleading wording. Text replaced by:

In contrast to $ToE_{rel}(T)$, most of the thermocline shows no emergence of the anthropogenic $O_2$ change by the end of the 21st century (Fig. 1d). In the remaining regions $ToE_{rel}(O_2)$ varies by about $\pm 40$ years.

**(5)**

Line 3 in the third paragraph in Section 3.3 on page 9. Large warming of more than $\sim 4$ Co is projected in the northern North Atlantic and round the subantarctic water. Can this projection be trusted?

Text modified to read:

Large warming of more than $4.0\pm0.7$ ° C is projected in the northern North Atlantic and around the subantarctic waters in the Indian and Atlantic Oceans (Fig. 4b and Fig. S4). We note that these projections are also characterised with the largest inter-model spread ($\pm 1.5$ ° C; Fig. 4d and Fig. S4) and uncertainties in these regional warming projections are large.

Many ESMs showed biases when simulating historical periods. Were these biases removed before the ToE analysis was applied?

No, we did not apply any bias corrections. Removing the bias in the climatological mean state would not affect the magnitude of the anthropogenic change ($S$) nor the variability ($N$). Therefore, all our results for ToE, $S$, and $N$ would be the same. Another potential approach might be to apply scaling factors to modelled variability and anthropogenic change. However, the observational information on industrial period change and on variability in T and $O_2$ in the thermocline is limited on the grid cell scale. This limitation regarding observational data may also apply to other more sophisticated correction methods.

**(6)**

Section 3.4. It was a good idea to check changes in AOU in order to better distinguish the O2 and temperature signals. Can you check if ventilation of the thermocline indeed decreases in regions with decrease in [-AOU] rather than relying on cited references?

The ideal age tracer was only provided for only one model family in the CMIP5 dataset (only GFDL-ESM2G and GFDL-ESM2M provide this tracer). This is stated in the manuscript and the corresponding text in section 2 reads: "Output of an ideal age tracer is not available for most models".

The authors were on the right track here to get at the mechanisms but did not go far enough. Similar mechanistic analysis should be done to explain the other results.

In the context of a multi-model study, it is challenging and difficult to develop robust mechanistic analyses for each single model as some of them do not provide all the variables necessary (for example in this case, the ideal age tracer).

**(7)**

Second paragraph in Section 4 (page 12). Most ESMs do not have fine resolutions to simulate the oxygen minimum zones (OMZs) well. As the authors indicated, the ESMs diverge in their projections for the physical and biogeochemical changes in OMZs. Some models even showed an opposite trend to the observations in recent decades.

This raised an important concern about the merit of even using such models to investigate ToE because they will lead to misleading results. Why didn't you remove those ESMs that did not capture the past changes?

We thank the reviewer for bringing up this issue. Nowadays, there is no generally accepted metric to weight the individual models regarding the simulated oxygen concentrations in the thermocline. This is due to the relative lack of observation. Cabré et al. (2015) show that all CMIP5 models present biases in $O_2$ concentration and in the extent of the $O_2$ minimum zone (e.g. see Fig.1 from Cabré et al., 2015). This is related to limited process understanding and coarse model resolution.

Moreover, Knutti and Sedláček (2013) underline the relatively high uncertainties in CMIP5 projections, and suggest the most reliable climate projection is given by a multi-model averaging. The limited number of models used in this study (only 4 different model family) is a caveat of our study that we discuss now in Sect 4 paragraph 2, which reads:

This multi-model study relies on results from only four different model families (GFDL-ESM, HadGEM2-CC, IPSL, MPI-ESM and CESM) applied in nine model configurations. All model configurations available from CMIP5 that provide 3-dimensional fields for $O_2$ and T for the control, historical and future-RCP8.5 scenario simulations have been incorporate into the analysis. Nevertheless, using a larger model ensemble would increase confidence in our results (Knutti and Sedláček, 2013).

**3 Anonymous Referee #3**

This paper focus on detecting the anthropogenic signals of both thermocline temperature and o2 (i.e. physical and biogeochemical properties) from a suite of CMIP5 models under the future projections. The study is (to some extent) based on the previous study of Hameau et al., (2019) extending to multi-model perspectives detecting the ToE of the thermocline temperature and o2 to assess the robustness of the results. The authors also introduce the relative ToE concept, results in reducing the inter-model spread compared to the traditional ToE and allows them to conduct more robust comparison. In general I think it is important to aim on understanding changes in both physical and biogeochemical tracers together to better understand the resulting changes in marine ecosystems and combining multi-tracers could provide additional insights. I think the topic and contents of this study fits into the scope of the special issue in Biogeosciences. However, I have comments on the current manuscript.

We thank the referee for the positive and constructive review.

**3.1 General comments**

**1. Abstract:**

I suggest to include some discussion (possibly in the section 4) on following up the statement "... the detection of anthropogenic impacts become more likely when using multi-tracer observations" in the abstract. Combination of two tracers will definitely provide additional information on further implications from both physical and biogeochemical perspectives. Despite the fact that two of these properties emerge on different timescales, what would authors expect to see from (or should be aware of for monitoring) future multi-tracer observations?

Multi-tracer observations would inform on the potential earlier impacts and consequences of the changing climate on marine ecosystems. Moreover, an important part of the biogeochemical processes simulated by climate models are still based on empirical values. Extended multi-tracer observations would contribute to better constrain climate models. The associated paragraph in the discussion has been extended and reads now:

Published studies addressing the detection of anthropogenic ocean warming focus on temperature at sea surface. To our knowledge, only a single study Hameau et al. (2019) using output from a single model is assessing ToE(T) in the thermocline. Yet, the thermocline is habitat for many fish and other species. Warming in combination with changes in other stressors, such as deoxygenation, ocean acidification and hypocapnia, may reduce the habitat suitability of marine life in a future climate (e.g. Deutsch et al., 2015; Gattuso et al., 2015; Breitburg et al., 2018; Cheung et al., 2018). Multi-tracer analyses contribute to a better understanding of the potential impact on marine ecosystems in a changing ocean.

[Figure]

Figure 3: Difference in $ToE_{rel}$ of T and $O_2$ for nine CMIP5 model configurations. The saturated colours indicate that one of the variables has not emerged by 2099.

**2.**

I agree that the ToE comparison among the models are not straightforward and the advantage of relative ToE is to "reduce the inclusion model uncertainty in the metric" (as stated in section 3.2 in details). However, in section 3.4 (results on ToE comparison between the two variables), the author calculate the difference between the "absolute ToE" for each models. I thought this will still include more model bias (from global ToE, which is subtracted in relative ToE). Since the author introduced an improved ToE metric, it might be better to come up with a metric comparing "relative ToE" from the two variables. This might not be straightforward but can you think of further metrics based on comparing two relative ToEs? If authors think the this will not make a difference, please explain in more details.

The goal of the metric relative ToE is to allow for a better comparison of ToE across models by removing potential biases (such as too high/low sensitivity to external forcing). We expect that these biases affect ToE(T) and $ToE(O_2)$ in a similar direction in an individual model. Therefore, the effect of biases may be reduced for the difference $\Delta ToE=ToE(T)-ToE(O_2)$, which is evaluated for each model and location individually. Further, and perhaps more important, the interpretation of the relative metric $\Delta ToE_{rel}=ToE_{rel}(T)-ToE_{rel}(O_2)$ is not very clear. $\Delta ToE$ indicates whether the $O_2$ signal or the T signal emerges first from the background variability, but this information is not provided anymore by $\Delta ToE_{rel}$. We evaluated $\Delta ToE_{rel}$ for each model for comparison with $\Delta ToE$. The results for $\Delta ToE$ (Fig. 6 of main manuscript) and $\Delta ToE_{rel}$ (Fig. 3 of this reply) are similar for individual models. In addition, the multi-model median of $\Delta ToE=ToE(T)-ToE(O_2)$ and of $\Delta ToE_{rel}$ are also very similar (not shown). Therefore, we do not discuss $\Delta ToE_{rel}$ in the manuscript.

**3.**

Regarding to the terminology used in the manuscript, the "internal natural variability" and "natural variability" are mixed used in the manuscript. It is not always clear what exactly the terminology defines in this context. From what I understand, the internal natural variability meant here is the variability stemming from internal climate system (specific example will be ENSO, PDO etc.) and the natural variability includes the natural "external" forcing (such as volcanic eruptions) correct? It was mixture of terminology (particularly in the discussion, which I saw the two terms were inter-changeable in some sentences) and I suggest to clearly define the terminology in the beginning and use the term in a consistent manner.

The reviewer points out an important potential source of confusion. The term "internal natural variability" has been replaced by the term "internal variability".

**4.**

I suggest the authors to explain some of the statistics in more details in the method. The noise (N) is defined as the standard deviations from the pre-industrial control simulations from each models but did the author defined standard deviations based on the temporal standard deviation using full control simulation period? This may not be a huge difference but I assume periods differs among the model. Also, the CESM1 in this study uses the last millennium spinup but is this different from the preindustrial control simulations (I assumed yes)? I think introducing a schematic based on for example Figure A1 c) d) (or similar figure) will help explaining the N and S, and at which point you define the ToE used in this study in a more visualized way.

We have modified the method section and added a Figure to the Appendix to clarify the points raised by the reviewers. Please see also our answers to reviewer # 1 and # 2 who raised similar issues. First, as the reviewer correctly points out, the preindustrial control simulations differ in their length: from 300 years (IPSL-CM5A-LR) to 1000 years (MPI-ESM-LR). We clarified this in the method section:

Although, the length of the control simulations differ between models, the entire duration of the control simulation is considered for each model to estimate the background noise.

Secondly, the reviewer correctly points out that the control simulation performed with the CESM1 was forced under 850 CE conditions, whereas CMIP5 control simulations are run under 1850 CE conditions. The following sentence was updated for more clarity:

The CMIP5 model simulations are branched off from preindustrial control simulations, whereas the CESM1.0 simulation is an extension of a last millennium simulation run under 850 CE conditions (Lehner et al., 2015)

Finally, we added a new figure to better illustrate the ToE definition (see Fig. A1)

**3.2   Specific Comments**

- Page 2, L24-25: I suggest to cite one of the Oschlies review paper in addition to Cocco and Bopp's papers (underestimating the trend and variability of o2 in the model simulations).

Reference to Oschlies et al. (2017) included as suggested. Acknowledged.

- Page 4, Method, Earth system models section: I am guessing this will not affect much on the overall results but why did you use your own CESM1 (with different spin-up procedures) for multi-model comparison instead of using CMIP5 CESM? In addition, is the CESM1 used in this study the same as the one in the early Hameau et al., (2019)?

The study considers all the available CMIP5 ESMs that provide oxygen, salinity and temperature 3-d fields for the piControl, historical and rcp85 simulations. Furthermore, the "in-house" CESM simulations that has already been described in Hameau et al., 2019 is used. The CESM simulation also follows the CMIP5 protocol (with the exception of the spin-up).

We have modified the corresponding text to:

In order to extent the multi-model ensemble from four to five family-models, we also included the output from simulations performed with the Community Earth System Model (CESM1.0) conducted at the Swiss Supercomputing Centre.

- Page 9, L1: What do you exactly mean by "combining climate sensitivity to anthropogenic forcing and natural variability in one metric"? I understand combining the anthropogenic forcing and natural variability part but I was not fully sure about the climate sensitivity statement.

We agree with the reviewer that this sentence might be confusing. We rewrote the sentence to:

The ToE allows for a comparison across climate models, by combining the amplitude of the climate response to anthropogenic forcing and the amplitude of natural variability in one metric.

- Page 12, L18: "an increasing ventilation" (following Gnanadesikan et al., 2007): Strictly speaking I would not state "an increasing ventilation" but it is more of a consequence of reduced upwelling as discussed in Gnanadesikan et al., 2007.

Thank you very much for this pointer. We agree with the reviewer and have modified the text as suggested.

- Figure 1. I understand from the Figure 1 that the SD reduces for the relative ToE but I also have some impression that two metrics could still give similar information. It might help to show additional map of ToE SD difference between Figure 1 (b) and (c) for example to show the bias (spread) reduction using this metric.

Thank you for this comment. We have replaced the lower panel (c, f) by the difference of the spread as suggested.

- Figure 6. For consistency, I suggest to used the same hatching as the previous figures to show the regions that one of the variables has not emerged by 2099 rather than saturated colors.

As suggested, hatching have been added in areas where both the T and $O_2$ signals do not emerge by the end of the 21st century. The caption of the figure has been updated with the following sentence:

No emergence in both T and $O_2$ are shown by the hatched areas.

- Figure 8. I like this summary figure aiming on incorporating emergent signals of both thermocline temperature and o2, along with AOU information (mainly indicating the water mass age information). Minor things on this, I think the x-axis is supposed to be -AOU (it puzzled me for a moment) and x-axis label should be corrected.

Many thanks for the positive comment. The label of the x-axis has been corrected as suggested.

**References**

[revised manuscript text omitted]

---

## Author Response (AR2)

**Reply to Referee**

We thank the reviewer for commenting this manuscript one more time. The original review comments are given below in black, our reply in blue, and quotes from the revised manuscript in gray.

**1 Anonymous Referee #3**

The authors did substantial revision and clarification following reviewers comments. I think now the aim of the study and the results are more clear and I really like the detailed explanation on methods section and schematic explanation in Figure A1. Despite the fact that ToE metric it self has numerous discussion, I think the multi-tracer ToE discussion in this study will be useful for the Large Ensemble Simulations and CMIP6 analysis.

At this point I only have minor (specific) comments before publication.
We thank the referee for the positive comments.

**1.1 Specific Comments**

-Section 3.2 (relative vs. absolute ToE): I would like to thank the authors for detailed reply and overall I understood the concept and advantage of relative ToE, basically allows to better compare the common patterns among the models because you remove the global mean bias. I wanted to further clarify but this means that the difference in relative ToE magnitudes among the models stem from mainly "regional model bias" correct (I would guess not everything)? I was still thinking of interpreting what causes the differences in relative ToE magnitudes among models and I would like further comments from the authors (and add one or two sentences in the main text if necessary).
Thank you for this comment. We have addressed this issue in our previous reply to referee 2.
We have not been able to find any obvious link between the multi-model spread of relative ToE and the multi-model median in ToE, or the multi-model median of the anthropogenic signal, or the multi-model median of the internal variability for both T and $O_2$ (see page 14, Fig. 2. of previous reply).
The following sentences have been added to section 3.1.1 second paragraph.
These regional differences in the multi-model spread could not be explained by the multi-model median of ToE, the anthropogenic signal nor the internal variability amplitude for both O2 and T. Scatter plots of individual grid cell values of the multi-model spread in ToE $_{rel}$ versus those of the multi-model median of ToE, the anthropogenic signal or the internal variability amplitude do not show a clear relationship (not shown).

-P10, L17-18: ... in accordance with (Levitus ... ), minor editorial thing but do you need parenthesis here?
The end of the sentence reads now:
...in accordance with published observational studies (Levitus et al., 2009, 2012; Bilbao et al., 2019).

-Abstract L5  P16, L19: The authors did an excellent work on revising and checking the consistency in terminology ("internal variability) and thank you for addressing this. These are additional details but are the "natural variability" in the abstract and P16 also suppose to be "internal variability" or did you intend to leave these as "natural variability"?
In the abstract, the "natural variability" term has been kept on purpose as we refer to naturally forced and

internal variability. The second mentioned occurrence is however corrected with internal variability and reads now:

For example, in the North Pacific subtropical gyre and the Southern Ocean, both the oxygen depletion and the internal variability are relatively strong.

5  -Figure 6, caption: From the caption sentence it is obvious what ToE is but I would still suggest to be more clear stating ToE = ToE(T)-ToE(O2) in the caption (if I did not miss, I also did not see what ToE is defined as in the main text).
The definition of $\Delta$ToE = ToE(T)-ToE(O$_2$) has been added to the caption.

10  -Figure 7, i), label: "CESM 1.0" -¿ Is this suppose to be "CESM 1.0-RCP8.5"?
Thank you for this remark. Figure 7 has been updated.

**References**

Bilbao, R. A. F., Gregory, J. M., Bouttes, N., Palmer, M. D., and Stott, P.: Attribution of ocean temperature change to anthropogenic and natural forcings using the temporal, vertical and geographical structure, Climate Dynamics, doi:10.1007/s00382-019-04910-1, URL https://doi.org/10.1007/s00382-019-04910-1, 2019.

Levitus, S., Antonov, J. I., Boyer, T. P., Locarnini, R. A., Garcia, H. E., and Mishonov, A. V.: Global ocean heat content 1955-2008 in light of recently revealed instrumentation problems, Geophysical Research Letters, 36, doi:10.1029/2008gl037155, 2009.

Levitus, S., Antonov, J. I., Boyer, T. P., Baranova, O. K., Garcia, H. E., Locarnini, R. A., Mishonov, A. V., Reagan, J. R., Seidov, D., Yarosh, E. S., and Zweng, M. M.: World ocean heat content and thermosteric sea level change (0–2000 m), 1955–2010, Geophysical Research Letters, 39, doi:10.1029/2012GL051106, URL https://agupubs.onlinelibrary.wiley.com/doi/abs/10.1029/2012GL051106, 2012.